# Epigenetic and parasitological parameters are modulated in *EBi3<sup>-/-</sup>* mice infected with *Schistosoma mansoni*

**Ester Alves Mota[1], Andressa Barban do Patrocínio○[2], Vanderlei Rodrigues[2], João Santana da Silva[2], Vanessa Carregaro Pereira[2], Renata Guerra-Sá○[1] ***

**1** Biochemistry and Molecular Biology Laboratory, Department of Biological Sciences, Universidade Federal de Ouro Preto, Campus Morro do Cruzeiro, Ouro Preto, Minas Gerais, Brazil, **2** Universidade de São Paulo, Medicine Faculty of Ribeirão Preto, Department of Biochemistry and Immunology; Vila Monte Alegre, Ribeirão Preto, São Paulo, Brazil

\* rguerrasa@gmail.com

**Data Availability Statement:** All relevant data are within the manuscript and its Supporting Information files.

## Abstract

*Schistosoma mansoni* adaptive success is related to regulation of replication, transcription and translation inside and outside the intermediate and definitive host. We hypothesize that *S. mansoni* alters its epigenetic state in response to the mammalian host immune system, reprogramming gene expression and altering the number of eggs. In response, a change in the DNA methylation profile of hepatocytes could occurs, modulating the extent of hepatic granuloma. To investigate this hypothesis, we used the *EBi3<sup>-/-</sup>* murine (*Mus musculus*) model of *S. mansoni* infection and evaluated changes in new and maintenance DNA methylation profiles in the liver after 55 days of infection. We evaluated expression of epigenetic genes and genes linked to histone deubiquitination in male and female *S. mansoni* worms. Comparing *TET* expression with *DNMT* expression indicated that DNA demethylation exceeds methylation in knockout infected and uninfected mice and in wild-type infected and uninfected mice. *S. mansoni* infection provokes activation of demethylation in *EBi3<sup>-/-</sup>*I mice (knockout infected). *EBi3<sup>-/-</sup>*C (knockout uninfected) mice present intrinsically higher DNA methylation than WTC (control uninfected) mice. *EBi3<sup>-/-</sup>*I mice show decreased hepatic damage considering volume and reduced number of granulomas compared to WTI mice; the absence of IL27 and IL35 pathways decreases the Th1 response resulting in minor liver damage. *S. mansoni* males and females recovered from *EBi3<sup>-/-</sup>*I mice have reduced expression of a deubiquitinating enzyme gene, orthologs of which target histones and affect chromatin state. *SmMBD* and *SmHDAC1* expression levels are downregulated in male and female parasites recovered from *EBi3<sup>-/-</sup>*, leading to epigenetic gene downregulation in *S. mansoni*. Changes to the immunological background thus induce epigenetic changes in hepatic tissues and alterations in *S. mansoni* gene expression, which attenuate liver symptoms in the acute phase of schistosomiasis.

**Funding:** Yes. Renata Guerra-Sá received the award. The grant numer is CNPq 310733/2014-6, Conselho Nacional de Pesquisa (full name funder), http://www.cnpq.br/. Ester Alves Mota received scholarship from CAPES (Coordenação de Aperfeiçoamento de Pessoal de Nível Superior) https://www.capes.gov.br. The funders had no role in study design, data collection and analysis, decision to publish, or preparation of the manuscript.

**Competing interests:** The authors have declared that no competing interests exist.

## Author summary

Schistosomiasis mansoni is a millennial disease and covers mainly tropical and economically disadvantaged regions. The disease mostly affects the liver, causing necrosis and fibrosis. Our study evaluated the effect of the immune system in modulating the oviposition of worms and the expression of genes mainly related to epigenetic regulation in worms. We also examined how methylation of liver DNA and the expression of related genes are influenced during *Schistosoma mansoni* infection depending on the type of immune response presented by the host. Our results revealed increased DNA demethylation during infection of *EBi3*[-/-] mice, while wild-type mice do not altered the methylation during infection. *S. mansoni* oviposition was decreased in infected *EBi3*[-/-] mice compared with wild-type mice, proving adaptation of the parasites to the type of host response. The livers of infected *EBi3*[-/-] mice presented less damage than those of wild-type mice, suggesting a protective hepatic effect conferred by the decrease in Th1 caused by IL27 and IL35 pathway knockout.

## Introduction

*Schistosoma mansoni* is a helminth parasite that causes schistosomiasis, a disease affecting about 200 million people in 56 countries. *S. mansoni* presents a heteroxenic life cycle, reproducing asexually inside *Biomphalaria* snails and sexually in mammals such as humans and rodents. Adult worms mate actively in the hepatic portal system and, subsequently, females will initiate egg production and release. Most of the eggs remain trapped in different organs of the definitive host, such as the liver, provoking inflammatory response and hepatic fibrosis. Nevertheless, some eggs reach the gut lumen and are liberated mingled with stool [1–3].

Oviposition, granuloma size, and cytokine and interleukin types can be modulated depending on the host profile, and variation in the aggressiveness of infection may be related to adaptation mechanisms of the parasite to the immunological environment in which it is found. Experimental infections in C57BL/6 mice (*Mus musculus*) generate immunological responses with greater intensity of interleukin IL10 and lower intensity of interleukins IL4 and IL13 when compared with infections in Balb/c mice [4]. Granulomas from rodent infections from natural reservoirs are less aggressive in the liver and more aggressive in the intestine; the composition of the granuloma in these animals is characterized by the presence of extracellular matrix and fewer inflammatory cells [5]. In contrast, Swiss strains infected with *S. mansoni* exhibit granulomas with extensive inflammatory infiltrates, often involving several eggs and the presence of necrosis [5].

Immunological approaches using *WSX*[-/-] mice (receptor chain of IL27) point to a reduction in IFN-γ levels compared with wild-type mice during acute and chronic stages of mansonic schistosomiasis but with no impact on liver damage [6]. EBi3 is a subunit of interleukin-27 (IL27); in this case combined with p28 forming a heterodimer, as well as interleukin-35, in which it also forms a heterodimer associating with p35. IL27 promotes the activation of Th1 by induction of IFN-γ expression, but induct T regulatory functions through IL10 signaling. IL35 activates regulatory T cells, acting in an opposite manner to the interleukins IL12 and IL23 of the same family, and is expressed only through molecules that promote the transcription of its gene, these two interleukins conducts regulatory functions mainly [7].

Epigenetics comprises changes in chromatin that do not involve changes in nucleotide sequence. These changes are triggered by environmental factors such as diet, social interaction, physical exercise, disease contagion, age, among others. Among the epigenetic alterations we

can mention modifications in the N-terminal portion of the histone tail (acetylation, deacetylation, ubiquitination, deubiquitination, methylation, demethylation, among others), DNA modifications, such as methylation and demethylation of cytosines, regulation by microRNAs and Long non-coding RNAs. These changes are fundamental for gene expression regulation, sex X chromosome silencing in females, transposon silencing [8,9]. Diseases that affect the liver causing collagen deposition induct alterations in the expression of DNA methyltransferases and DNA demethylases, leading to disturbance in expression of several genes [10,11], here we intended evaluate in schistosomiasis. *DNMT1* enzyme is related to the maintenance of existing methylation during the replication process, copying the patterns from the template to the new strand, and is more expressed in the period before mitosis. DNMT3A and B are enzymes that lead by stimulus create new methylation patterns in DNA, they are *de novo* methyltransferases [12]. TET1, 2 and 3 act synergistically and procedurally by removing 5-mC, the first step for 5-hydroxymethylcytosine (5-hmC), and following 5-formylcytosine (5-fC), 5-carboxycytosine (5-caC) and then return to usual cytosine [13].

As different intensities of interleukins are detected during schistosomiasis depending on mouse lineage, immunological profile might induce alterations in the signaling pathways of worms allowing adaptation to the host. The presence of different epigenetic marks and changes in the patterns of these marks are involved in the transition of the evolutionary stages of *S. mansoni* and host–parasite interaction; H3K4me3 and H3K27me3 show greater variation and influence in the transformation of the parasite among phases [14,15]. Epigenator molecules native to intracellular parasites such as protozoa trigger into host new metabolic pathways or interfere inducing initiator host proteins to modify metabolism and promote an environment suitable for parasite reproduction [16]. Initiator host proteins rely mainly on alteration in transcription of cytokines, chemokines, proteins related to cell proliferation and apoptosis, and changes in the immunological setting [16], and the intracellular parasite *Leishmania donovani* has been observed to induce methylation in host macrophage DNA [17].

Aiming highlight some impact of immune system on epigenetic of *S. mansoni*, we explored the gene expression involved in chromatin modifications. Among these genes, *SmGCN5* is a histone acetyltransferase that targets histone N-terminal tail lysine residues. *S. mansoni* ortholog presents HAT and bromodomain, which also occurs in *Homo sapiens* and other species orthologs compared by the authors, *SmGCN5* associates with nuclear receptors, and has high expression rates throughout *S. mansoni* life cycle [18]. HDACs enzymes remove the acetyl group from histones, conducting to gene silencing; *S. mansoni* has HDAC1, 3 and 8 and are expressed at all stages of the life cycle. HDAC8 has amino acid insertions that differ from HDACs present in the definitive host, so it has already been proposed as a target for inhibitors to implement new treatments in the future, according to these reports, HDAC8 shows both nuclear and cytoplasmic cellular localization [19]. MBD methyl CpG binding domain is a protein that binds to methylated cytosines, and *S. mansoni* expressed it at all stages of the parasite's life cycle including both sexes. In the same approach the authors found that this protein has nuclear localization only and relates to oviposition [20]. Previous studies identified 17 USP orthologs in *S. mansoni*, by homology with 56 USPs present in *H. sapiens*, it was detected that the expression of these USPs is altered in stages of egg, cercariae, schistosomula and adult worms, reflecting a recycling of proteins used by the parasite during the evolutionary stage transition [21]. USPs can prevent different forms of degradation, i.e. proteasome [9], lysosomal [22], that are signaled by lysine ubiquitination. USPs also act on transcriptional regulation, since many USPs target histones, ubiquitinated histones may indicate chromatin activation or repression depending on histone variant and K residue [9]. Substrate deubiquitination competes with ubiquitination and can reverse substrate fate and recycle polyubiquitinated conjugate chains. The position of the lysine to be ubiquitinated is directly related to the type of

signaling generated [9]. In this work USPs 7, 15, 22, 46 and 49 were selected among others because they have histones as a common target, potential epigenetic function in *S. mansoni*.

We hypothesize that *S. mansoni* reprogram gene expression in response to the immune system of the mammalian host, in this case C57BL/6 *EBi3-/-* and alters the number of eggs produced. In response, mammalian host alters DNA methylation profile of hepatocytes by changing expression levels of DNMTs, TETs and the percentage of 5-mC and also modulates the extent of hepatic granuloma. Results presented in this work indicate that some parasitological parameters such as: number of eggs per gram of liver, number of hepatic granulomas, granuloma volume and number of eggs per gram of feces, are altered in the infected knockout model leading to epigenetic changes in mice and worms. To investigate the epigenetic interactions between *S. mansoni* and the mammalian host, the present work consists of three strands: (1˚) the investigation of the parasitological effects of the C57BL/6 *EBi3-/-* model (quantification of eggs trapped on tissues and feces and parasites); (2˚) if the parasite would modulate epigenetic mechanisms in the host (expression analysis of DNMT and TETs, % 5-mC and miRNAs in liver) and (3˚) if the host could affect the expression of genes related to parasite epigenetic mechanisms (expression of several genes related to epigenetic modifications in *S. mansoni* and some miRNAs).

## Results

### *EBi3-/-* influence on parasitological parameters

*S. mansoni* parasite load in experimentally infected wild-type mice and *EBi3-/-* mice was estimated by quantifying adult worms recovered by perfusion of the hepatic portal system and by counting eggs retained in the intestine, liver and feces. A mean of 480.9 ± 107.5 eggs per gram of intestine were quantified in *EBi3-/-*I mice compared with 526.3 ± 72.26 eggs per gram of intestine in WTI mice (Fig 1A). Although the number of eggs aggregated in the intestines of WTI mice was greater than that in *EBi3-/-*I mice, this difference was not significant. By contrast, *EBi3-/-*I mice had significantly fewer eggs aggregated in the liver than WTI mice (2735 ± 413.6 vs. 8178 ± 1310 eggs per gram of liver, respectively, p = 0.0029) (Fig 1B).

A total of 100 cercariae were inoculated per mouse; a mean of 19.96 ± 2.051 worms were recovered from *EBi3-/-* mice and 19.33 ± 2.726 from WTI mice. No statistical difference was observed in the recovery of worms between experimental groups (Fig 1C). Coupled worms were also quantified, but no difference was detected between *EBi3-/-*I (4.826 ± 0.8906) and WTI (6.833 ± 1.256) mice (Fig 1D).

The number of granulomas in livers of WTI mice (median = 57.5) was significantly higher than that in *EBi3-/-*I mice (median = 22), with p = 0.0001 (Fig 1E). Granuloma volume was also higher in WTI mice than in *EBi3-/-*I mice, with p < 0.0001 (Fig 1F). Hepatic damage was greater in the presence of the immune response involving EBi3. Comparing the appearance of WTI and *EBi3-/-*I granulomas revealed that granulomas in knockout mice were frequently diffuse and disorganized. In WTI animals, granulomas were concentric, with greater presence of extracellular matrix and abundant fusions between granulomas (see microscopy in S1 Fig). Egg numbers in feces were augmented significantly in WTI mice compared with *EBi3-/-*I mice (135.8 ± 9.181 vs. 17.76 ± 5.938 eggs per gram, p<0.0001) (Fig 1G).

### Changes in *de novo* DNA methyltransferase and DNA demethylase expression in *EBi3-/-* liver linked to *S. mansoni* infection evidenced by qRT-PCR

In this work we evaluated the expression of maintenance and de novo DNA methyltransferases, as well as DNA demethylases in liver mice, in order to detect if the immune

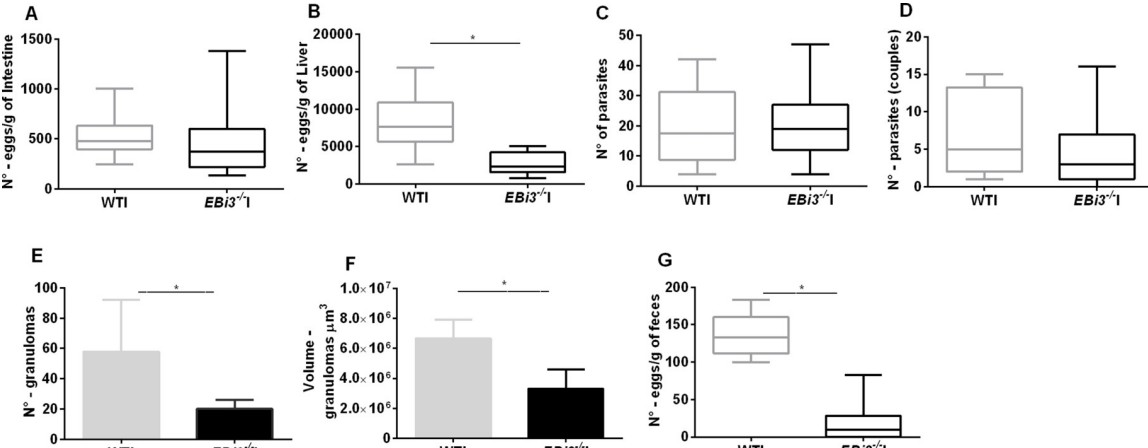

**Fig 1. Parasitological parameters of infected *EBi3<sup>-/-</sup>* and WT mice.** All mice were males, they had 17 g and 5 weeks old when were infected. The mean or median profile, depending on the test used, for 10 WTI mice and 15 *EBi3<sup>-/-</sup>*I mice of the first perfusion are represented, all statistical analysis used * p < 0.05. (**A**) Number of eggs recovered from the gut by digestion with KOH; (**B**) number of eggs recovered from the liver, p = 0.0029; (**C**) number of individual parasites recovered by perfusion of the hepatic portal system; (**D**) number of worm couples recovered; *t* test, * p < 0.05. (**E**) Number of hepatic granulomas found in 20 fields at ×20 magnification; Mann–Whitney test, p < 0.0001. (**F**) Volume of granulomas; *t* test, p = 0.0001. (**G**) Number of eggs recovered from feces; *t* test, p < 0.0001; GraphPad Prism 6. Error bars show the standard deviation.

background would affect the expression of these enzymes and consequently the methylation patterns in DNA conducting to gene repression or activation. Gene expression was quantified using the *HPRT1* gene as calibrator, there was no statistical difference in *DNMT1* expression among WTC, WTI, *EBi3<sup>-/-</sup>*C and *EBi3<sup>-/-</sup>*I mice, with low levels of expression in all groups (Fig 2A). WTC and WTI showed differences in *DNMT3A* expression, however, which was lost in the knockout model (Fig 2B).

A similar pattern was observed for *TET1* demethylase, with a significance difference in expression only observed between WTC and WTI (Fig 2D). Expression levels of the *DNMT3B* gene were affected by *S. mansoni* infection in knockout mice, as evidenced by differences in expression between *EBi3<sup>-/-</sup>*C and *EBi3<sup>-/-</sup>*I (Fig 2C). There was a statistically significant decline in *DNMT3B* expression in *EBi3<sup>-/-</sup>*I compared to WTI. Expression levels of *TET2* showed differences between the wild-type control and the knockout control, but differences in demethylase expression did not prevail after *S. mansoni* infection (Fig 2E). The *TET3* demethylase gene showed the highest levels of expression, with the difference between the controls also observed among infected mice, indicating little impact of *S. mansoni* infection on expression of this gene (Fig 2F).

Correlation analysis among the expression of *TET1*, *2* and *3* and *DNMT1*, *3A* and *3B* indicated that there was a negative correlation between *DNMT1* and *TET3* gene expression and granuloma number in *EBi3<sup>-/-</sup>*I (Fig 2G–2H). Liver damage induced downregulation of *DNMT1* and *TET3* expression considering granulomas number. Expression levels of the *DNMT3A*, *B* and *TET1* and *2* genes did correlate with number and volume of granulomas.

Demethylation was more active than DNA methylation in wild-type and knockout mice considering the relative expression levels of TET and DNMT genes. The expression levels of *DNMT1* were different from those of all demethylases, while the expression levels of *TET2* and *TET3* were different from those of each DNA methyltransferase in control wild-type mice, as shown in S2A Fig. *S. mansoni* infection produced maintenance of *TET1* and *DNMT1* expression levels and upregulation of *DNMT3A* and *B* expression levels in wild-type mice, resulting in levels similar to those of *TET2* and *3*; schistosomiasis induced high levels of methylation in infected wild-type mice compared with control wild-type mice as shown in S2B Fig. The DNA

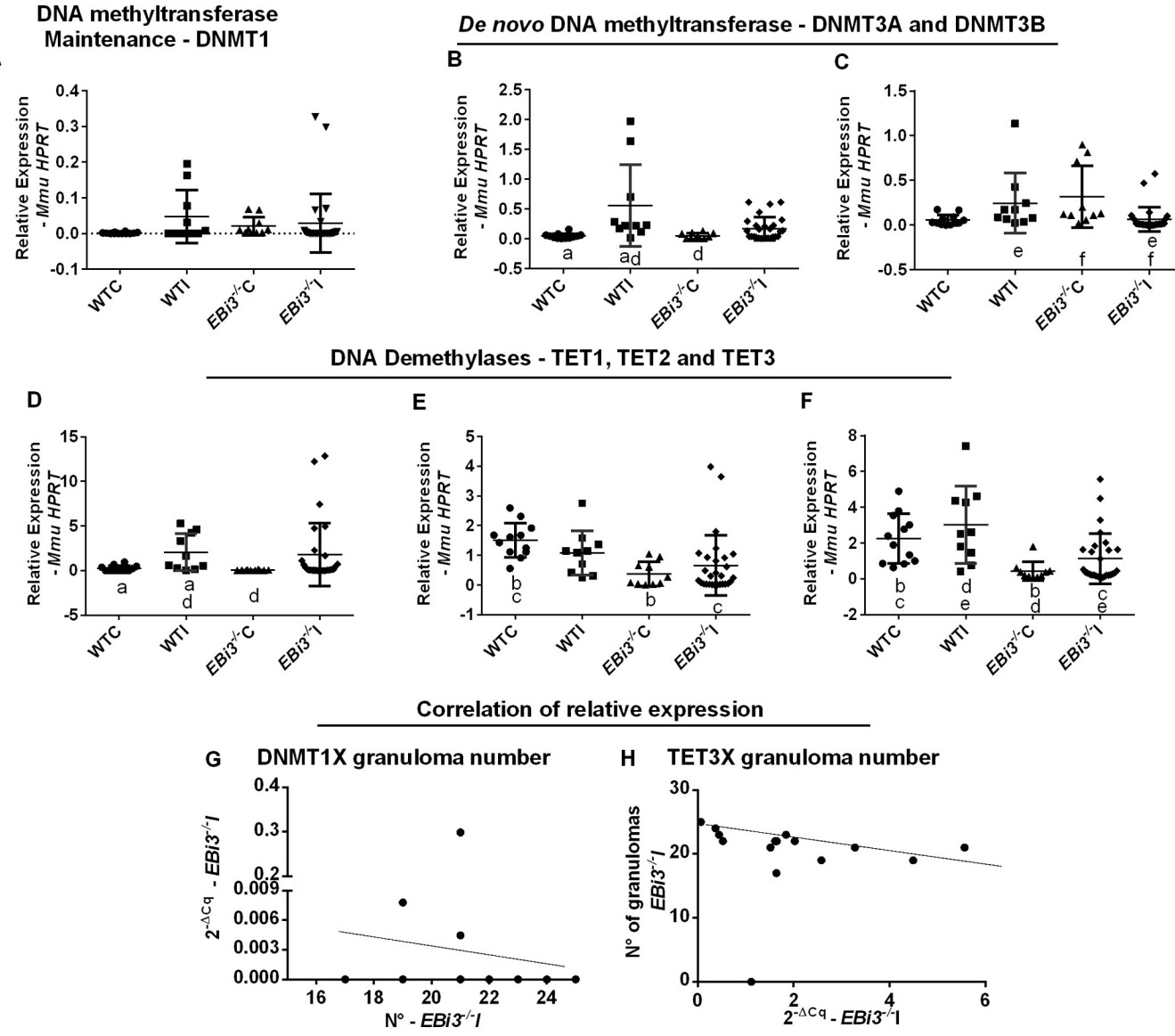

**Fig 2. Relative expression (qRT-PCR) of DNA methyltransferases and demethylases. (A)** *DNMT1*, p = 0.0348 **(B)** *DNMT3A*, p = 0.0008 **(C)** *DNMT3B*, p = 0.0001 **(D)** *TET1*, p = 0.0041 **(E)** *TET2*, p = 0.0001 **(F)** *TET3*, p = 0.0001, in wild-type control (WTC n = 12) and infected mice (WTI n = 10), and control (*EBi3⁻/⁻*C n = 10) and infected (*EBi3⁻/⁻* I n = 28) knockout mice. Kruskal–Wallis, p < 0.05; GraphPad Prism6. Comparisons indicated by the following letters are significant. a: WTC vs. WTI; b: WTC vs. *EBi3⁻/⁻*C; c: WTC vs. *EBi3⁻/⁻*I; d: WTI vs. *EBi3⁻/⁻*C; e: WTI vs. *EBi3⁻/⁻*I; f: *EBi3⁻/⁻*C vs. *EBi3⁻/⁻*I. **(G–H)** Spearman correlations: gene expression level for each *EBi3⁻/⁻*I mouse was correlated with granuloma number for the same mouse, all correlations were tested using p < 0.05. **(G)** *DNMT1* expression level and granuloma number (p < 0.0001; r = -0.3302); **(H)** *TET3* expression level and granuloma number (p = 0.0415; r = -0.475). Error bars represents standard deviation. Dots in all graphs represent a mouse, totalizing 60 mice of all groups.

methyltransferase transcriptional activation provoked by *S. mansoni* in wild-type mice was reversed in *EBi3⁻/⁻* I mice; expression levels of *TET1*, *2* and *3* were raised while those of DNMTs dropped substantially in *EBi3⁻/⁻* I mice compared with those in *EBi3⁻/⁻* C mice. *S. mansoni* induced differences between *DNMT1* and *3A* expression levels in *EBi3⁻/⁻* I livers, as shown in S2C and S2D Fig.

## Schistosomiasis increases % 5-mC of hepatic DNA in *EBi3<sup>-/-</sup>* mice

In DNA, 5-methylcytosine content quantified by Elisa assays using anti-5-methylcytosine revealed methylation contents of at least 0.5% of 5-mC. Knockout mice had statistically naturally higher percentages of % 5-mC in relation to wild-type mice in the liver (*EBi3<sup>-/-</sup>*C vs. WTC, $p < 0.05$) *EBi3<sup>-/-</sup>*C had 1.58 times more of % 5-mC than WTC; this difference was lost (*EBi3<sup>-/-</sup>*I vs. WTI, $p < 0.05$). *S. mansoni* infection reduced % 5-mC of hepatic DNA rates significantly in *EBi3<sup>-/-</sup>*I mice ($p<0.05$) but not in WTI mice (Fig 3), *EBi3<sup>-/-</sup>*C mice contained 1.96 times more of % 5-mC than *EBi3<sup>-/-</sup>*I. The reduced levels of % 5-mC DNA in livers of *EBi3<sup>-/-</sup>*I mice compared to *EBi3<sup>-/-</sup>*C mice (Fig 3) was supported by the increase in *TET* expression levels presented in S2D Fig.

## *S. mansoni* miRNAs are modulated by the mouse immune system

MiRNAs important to liver metabolism as miR-122 and to fibrosis process as miR-31 were tested in this work to verify possible alterations caused by immune response coupled with infection. No statistical difference was observed in the fold change of miR-122-3p, miR-122-5p, or miR-31-3p and miR-31-5p levels between infected and control C57BL/6 *EBi3<sup>-/-</sup>*. This suggests that the liver damage caused in knockout mice may not produce alterations in release of this miRNAs (Fig 4A–4D).

MiRNAs from *S. mansoni* were analyzed also to verify if the mouse immune system would affect the expression of these regulatory molecules in the parasite. Males and females worms recovered from WTI were used as control group and males and females worms recovered from *EBi3<sup>-/-</sup>*I were used as test group. Male and female *S. mansoni* worms taken from knockout mice expressed less miR-124-3p (Fig 4E) than those retrieved from wild-type mice. miR-125A and miR-125-3p showed expression only in females (Fig 4F). Males from knockout mice had higher expression levels of miR-190-3p and miR-190-5p (Fig 4G and 4H) than males recovered from wild-type mice; however, in females the opposite occurred, with females recovered from wild-type animals showing higher expression of miR-190-3p and 5p than females recovered from knockout animals. miR-190-3p and 5p probably have sex-specific functions because their expression levels in *EBi3<sup>-/-</sup>* mice showed statistically significant differences between males and females.

Spearman matrix correlation compared the parameters, (such as parasitological, DNA 5-mC %, expression of some genes and miRNAs, among others) of each mouse to each other to observe the influence of the parameters analyzed over them. The expression parameter of *TET2* correlated best with the others in the WTC group, considering $p <0.05\%$. TET2 was positively correlated with *TET3*, miR31-3p and miR122-3p. In addition, TET3 expression correlated positively with DNMT3A expression and miR31-3p also positively correlated with miR122-3p, as shown in S1 Table.

In WTI group, *DNMT1* expression correlated positively with eggs in feces, miR122-5p expression correlated negatively with *DNMT3B* and positively with number of eggs in the intestine. Also, eggs in the liver and intestine correlated positively with each other, as shown in S1 Table.

*EBi3<sup>-/-</sup>*C group presented the lowest number of correlations between the parameters considering the four groups analyzed. In this group *TET2* expression correlated positively with TET3 expression. *DNMT1* expression correlated positively with the liver's DNA 5-mC % content and miR122-3p expression as shown in S1 Table. *EBi3<sup>-/-</sup>*I group was the group that presented the highest number of correlations between the parameters. The *TET1* gene expression correlated negatively with DNMT1 gene expression. TET3 correlated positively with TET2, DNMT1, and negatively correlated with granulomas number and miR31-3p and 122-3p

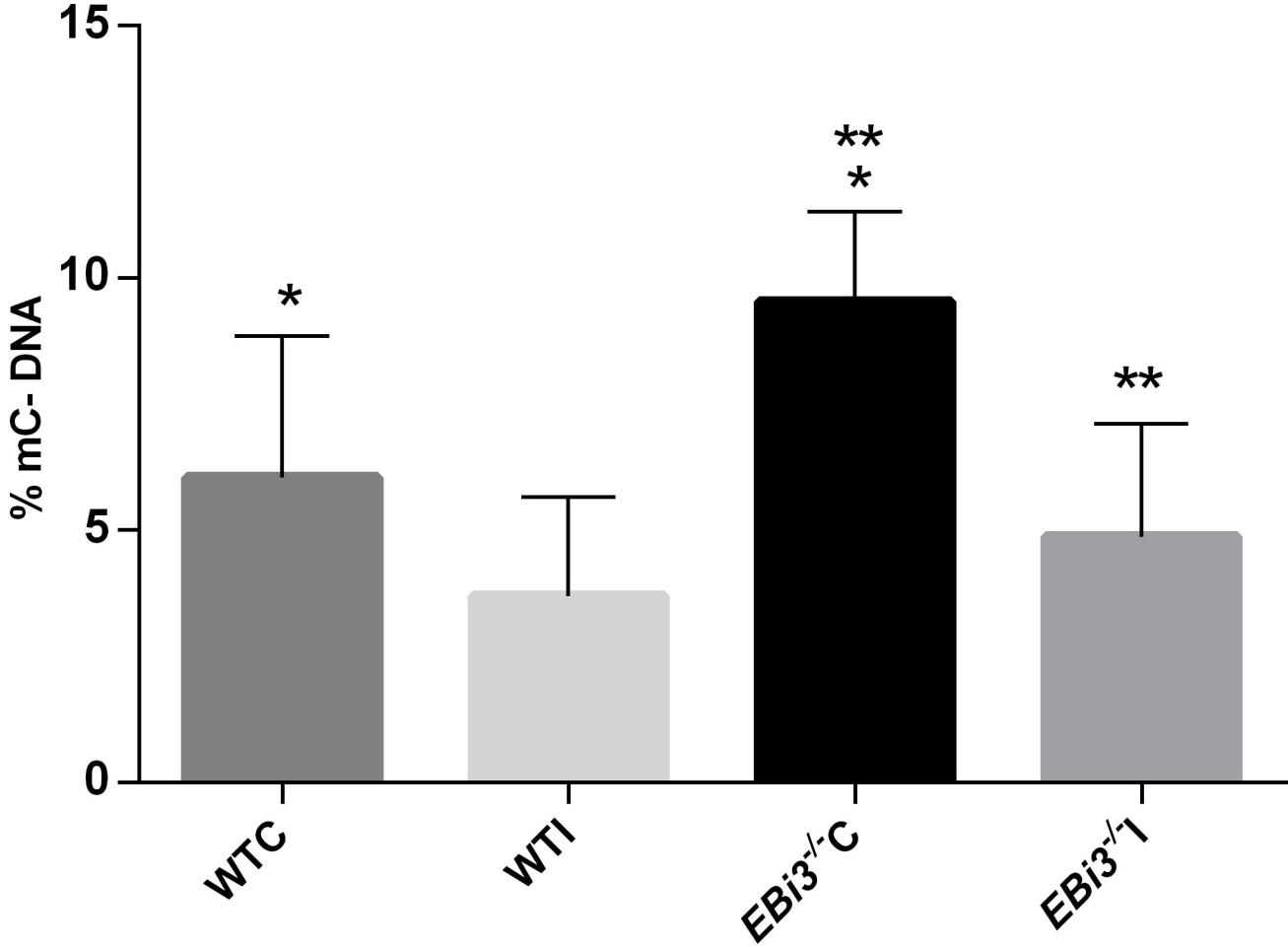

**Fig 3. Methylation content of mouse liver DNA (% 5-mC).** The figure represents the percentage of DNA 5-mC for the first perfusion mice. Error bars represents standard deviation of WTC (n = 12), WTI (n = 10), *EBi3[-/-]*C (n = 5) and *EBi3[-/-]*I (n = 15) groups, each mouse had a technical duplicate of the biological sample using One-way ANOVA, $p < 0.05$. *: WTC vs. *EBi3[-/-]*C; **: *EBi3[-/-]*C vs. *EBi3[-/-]*I; $p = 0.0004$.

expression. DNMT1 expression correlated positively with eggs in feces and negatively correlated with miR31-5p and 122-3p expression. Granulomas volume correlated positively with the number of eggs in the feces. The number of eggs in the intestine correlated positively with the number of granulomas found in the liver. MiR122-3p correlated positively with the number of granulomas found in the liver and miR122-5p expression as shown in S1 Table.

### Livers of infected *EBi3[-/-]* mice harbor native *Schistosoma* lncRNAs

All lncRNAs presented in Table 1 were quantified by their relative expression in the livers of infected and uninfected mice. These lncRNAs are found in *S. mansoni*, and detecting their presence in the livers of infected mice may aid in the development of new biomarkers for schistosomiasis. *LncRNAs 5, 6, 8* and *9* (Fig 5) (neighboring beta-proteasome subunit 1, U3 nucleolar ribonucleoprotein and growth factor receptor of fibroblasts, actin suppressor (sac) and rRNA) had the highest relative expression in *EBi3[-/-]*I mice [23]. LncRNAs not shown in Fig 5 were not expressed in the livers of infected mice or were expressed in both infected and uninfected mice.

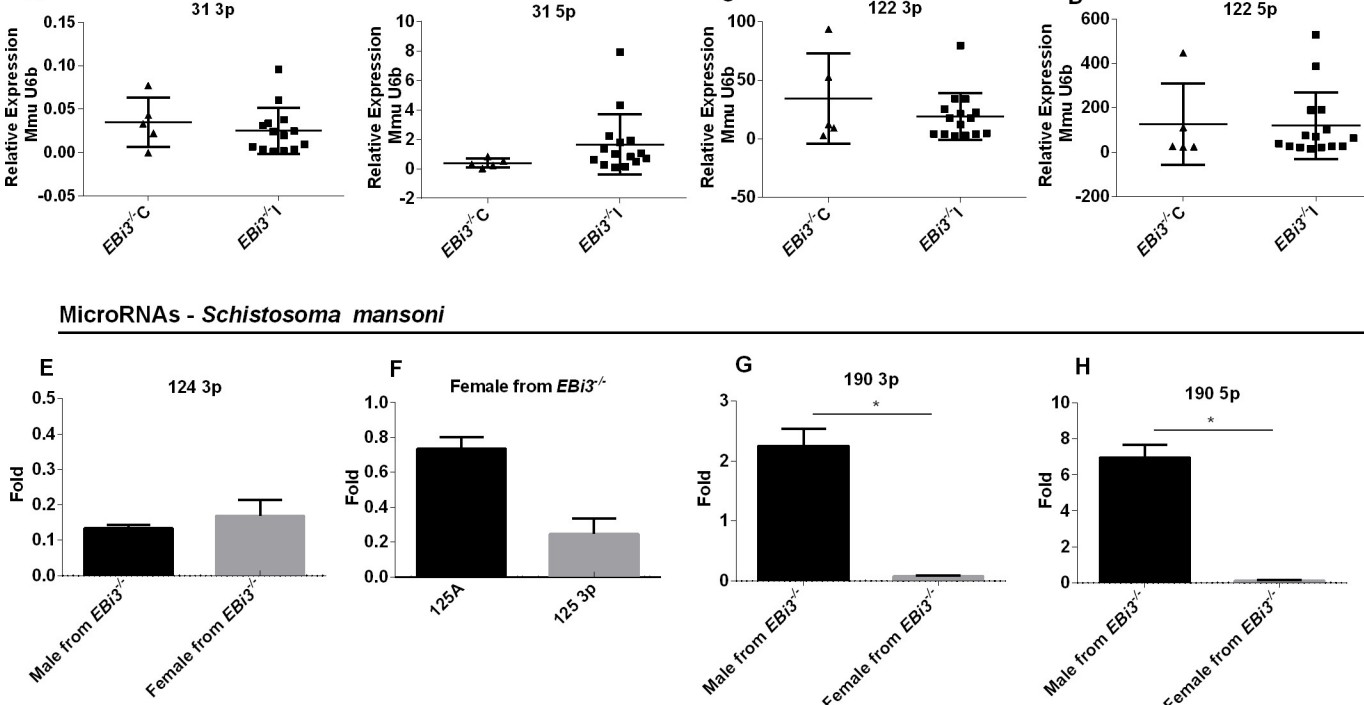

**Fig 4. Relative expression of miRNAs.** Relative expression of (**A**) miR-31-3p, (**B**) miR-31-5p, (**C**) miR-122-3p and (**D**) miR-122-5p levels between livers of infected and uninfected C57BL/6 *EBi3⁻/⁻* mice based on RT-qPCR. Small nuclear RNA U6 (SnU6) was used for calibration. Mann–Whitney U tests were performed; *p < 0.05. Dots in graphs relative to mice miRNAs represent a individual mouse (**E–H**) Fold change of miRNAs in *S. mansoni* males and females recovered from infected and uninfected C57BL/6 *EBi3⁻/⁻* mice and infected and uninfected C57BL/6 WT mice based on RT-qPCR and Unpaired T with Welch's correction, *p < 0.01 were performed. Error bars represents standard deviation: (**E**) miR-124-3p, (**F**) miR-125-3p and miR125A, (**G**) miR-190-3p (p = 0.0055) and (**H**) miR-190-5p (p = 0.0033).

The analysis of the presence of *S. mansoni* lncRNAs in *EBi3⁻/⁻*I mouse liver was made in order to select very preliminary possible biomarkers for schistosomiasis, if they did not also express in the control group. In this experiment samples from 5 *EBi3⁻/⁻*C and 13 *EBi3⁻/⁻*I were used. But to confirm lncRNAs as biomarkers, their detection in serum is necessary. Further experiments are needed to confirm that these lncRNAs are specific to *S. mansoni*.

## Sets of *S. mansoni* genes disturbed by the host immune system

This work aimed also to detect whether genes related to epigenetic changes in *S. mansoni* are differentially regulated in parasites that developed in a different immune environment (wild-type versus knockout mouse). Genes encoding the histone deacetylase *SmHDAC3* and ubiquitin-specific proteases *SmUSP15* and *SmUSP49/44* showed no statistical difference in expression levels among the four groups, as shown in S3 Fig. Sex-specific expression of the *SmMBD*, *SmUSP7*, *SmUSP22* and *SmHDAC8* genes was lost in worms isolated from *EBi3⁻/⁻* mice; by contrast, expression of *SmUSP46* was maintained and *SmHDAC1* gained sex-specific expression compared with worms isolated from WT mice (Fig 6). Male parasites recovered from knockout mice showed downregulation of *SmMBD*, and *SmUSP7*, *22* and *46* gene expression, whereas only expression of the *SmUSP7* gene was downregulated in the female parasites compared with those recovered from WT mice. Worms recovered from knockout mice only showed upregulation of *SmHDAC1* expression in females. Male parasites recovered from WT mice showed high expression levels of *SmGCN5* compared with female recovered from *EBi3⁻/⁻*

**Table 1. Oligonucleotides used to evaluate *S. mansoni* gene expression.**

| Oligonucleotide | Sequence | Source |
|---|---|---|
| ***SmMBD*** Gene DB Smp_138180 | F: 5'-CGGTTATTGTACGTTCTCATCC-3' <br> R: 5'-CAGCACTCTGTATTCCTTTAGGC-3' | [51] |
| ***SmGCN5*** Gene DB (Smp_070190) | F: 5'-GTGCAATGCAAATGGTTGAC-3' <br> R: 5'-TTACGAAGGCGCTTCAAAAT-3' | [18] |
| ***SmHDAC1*** Gene DB (Smp_005210) | F: 5'-GCGAGTATTTTCCCGGAACTG-3' <br> R: 5'-CGCGACCTGCACCAATATCT-3' | [19] |
| ***SmHDAC3*** Gene DB (Smp_093280) | F: 5'-TTTTACGATCCAGATTGTGGGA-3' <br> R: 5'-CATGGGATGATTCGGTCCATA-3' | [19] |
| ***SmHDAC8*** Gene DB (Smp_091990) | F: 5'-GGCATCAATGATTTGGACTGG-3' <br> R: 5'-TCGGCTCCGCATTGAACTAC-3' | [19] |
| ***SmUSP 7*** Gene DB (Smp_089180) | F: 5'-TTATGTCTCACCCGTCTC-3' <br> R: 5'-CCCACTCTCTACAGAACAG-3' | [21] |
| ***SmUSP 15*** Gene DB (Smp_128770) | F: 5'-TAGAACAAGAGCGTCCAC-3' <br> R: 5'-GCAATCAGGGAGGCATAC-3' | [21] |
| ***SmUSP 22*** Gene DB (Smp_074400) | F: 5'-CTGTAAATGGCTGCTCTG-3' <br> R: 5'-ACGGGTGTATGGGTCAAC-3' | [21] |
| ***SmUSP 46*** Gene DB (Smp_000710) | F: 5'-TCGGAGTAGATGCTGAAG-3' <br> R: 5'-GGCACCTAGTTTCATTGG-3' | [21] |
| ***SmUSP 49/44*** Gene DB (Smp_123630) | F: 5'-GGATTGGTGTGTTGTTCTTC-3' <br> R: 5'-CGCATCTCCATCTTGTAG-3' | [21] |
| ***SmEIF4E*** | F: 5'-TGTTCCAACCACGGTCTCG-3' <br> R: 5'-TCGCCTTCCAATGCTTAGG-3' | [52] |
| ***Sm-lncRNA 1*** | F: 5'-AAGGGATGAGTTGACTGC-3' <br> R: 5'-ACACGAAGACACCTATGACC-3' | [23] |
| ***Sm-lncRNA 2*** | F: 5'-AGACAATGCGATGCCGTTAG-3' <br> R: 5'-TTTGGAACTCGTCAGCTAGG-3' | [23] |
| ***Sm-lncRNA 3*** | F: 5'-TCACATCTCGCAACTCAG-3' <br> R: 5'-AGTGGTCGTCAAGCAAAC-3' | [23] |
| ***Sm-lncRNA 4*** | F: 5'-TTTCGACACGGCAACTGATC-3' <br> R: 5'-GCCGATTCAGTGTAGCAAAG-3' | [23] |
| ***Sm-lncRNA 5*** | F: 5'-GATCGAGCTGTAACTGCAC-3' <br> R: 5'-GATCCACATCCATATGAGTG-3' | [23] |
| ***Sm-lncRNA 6*** | F: 5'-GACTGTTGGAAGAGGAAATG-3' <br> R: 5'-GAGGATTTAAGCGACCATTG-3' | [23] |
| ***Sm-lncRNA 7*** | F: 5'-CCGATGAGATGCGTATAG-3' <br> R: 5'-GCAACACAGTGAGGTAGAG-3' | [23] |
| ***Sm-lncRNA 8*** | F: 5'-CCACACAGGTAGTTCAGC-3' <br> R: 5'-GAATCACTTGCACTTCGC-3' | [23] |
| ***Sm-lncRNA 9*** | F: 5'-CTGTGAGAATGGTGGATG-3' <br> R: 5'-ACGTTTATGAGCCGTAGC-3' | [23] |
| ***Sm-lncRNA 10*** | F: 5'-GTGATATGCCCGGACAAAG-3' <br> R: 5'-TTGAACGAGCAGCTGGAC-3' | [23] |
| ***Sm-lncRNA 11*** | F: 5'-CCTCGTGTTTGTGCTTTG-3' <br> R: 5'-GGAATGTGATTGCCTAGTCG-3' | [23] |
| ***Sm-lncRNA 12*** | F: 5'-GCACTTGACACTAACCAGG-3' <br> R: 5'-GGAGCTGTTCACTCATTG-3' | [23] |
| ***Sm-lncRNA 13*** | F: 5'-TTCCCTCCAGACTATGATCC-3' <br> R: 5'-CACGTATTGCACCTGATG-3' | [23] |
| ***Sm-lncRNA 14*** | F: 5'-GTTGAAGAAGGTGAGTGC-3' <br> R: 5'-GTGGAGGACTTGGAGATAC-3' | [23] |
| ***Sm-lncRNA 15*** | F: 5'-CCATGCAAGTGTGATCCG-3' <br> R: 5'-GTGGGATTATCAGCTGCAGG-3' | [23] |
| **LTR- Retrotransposon Saci 4** | F: 5'-GGGTGCATCAGAGTAATC-3' <br> R: 5'-ACTTGATCCGCATACTCC-3' | [23] |

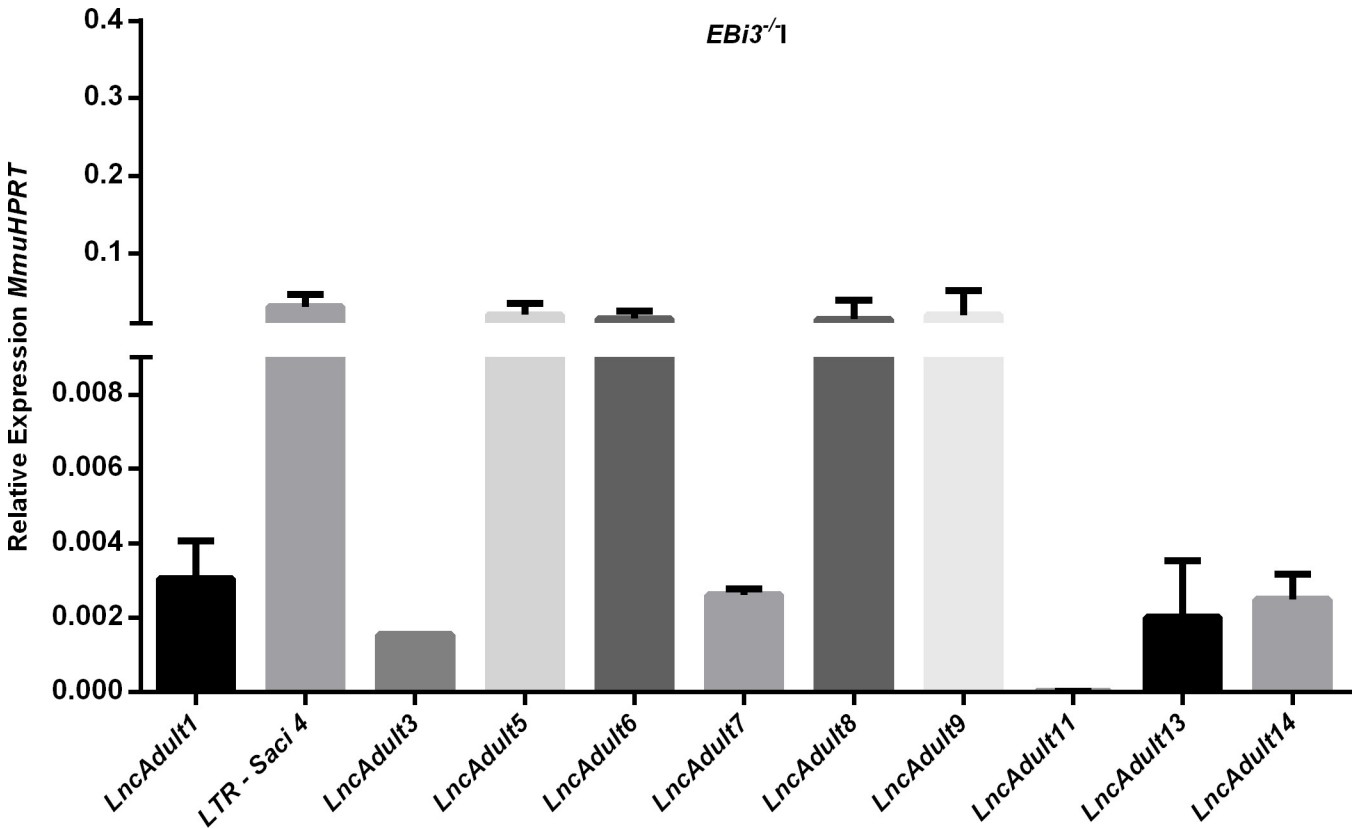

**Fig 5. Relative expression of *S. mansoni* lncRNAs in the livers of C57BL/6 *EBi3⁻/⁻*I mice by RT-qPCR.** All lncRNAs detected in schistosomula, cercariae and eggs were also evaluated in the liver, but lncRNAs with very low or undetectable expression levels are not shown in the figure. *Mus musculus* (mmu)-hypoxanthine-guanine phosphoribosyltransferase (*HPRT*) was used for calibration. Error bars show the standard deviation.

mice. Absence of the IL27 and IL35 pathways in *EBi3⁻/⁻* knockout mice influenced both male and female worms to lower levels of transcription.

Analysis of the relative expression of *SmUSP7* to the calibrator gene *EIF4E* revealed that there is a difference between male and female controls, the male control worms is 3.4 times upregulated compared to the female control worms. There is a difference between the two groups of males, the recovered *EBi3⁻/⁻*I male worm is 6 times downregulated compared to the male control worms and between the two groups of females, the recovered *EBi3⁻/⁻*I female worm is 5 times downregulated compared to the female control worms.

Regarding the expression of *SmUSP22*, the male control worms is 4.6 times upregulated compared to the female control worms. The male recovered from *EBi3⁻/⁻*I is 7.2 times downregulated compared to the male control worms.

The USP gene that was most influenced by the host immune system in *S. mansoni* was *SmUSP46*. It was observed that male control worms were 2.6 times upregulated compared to female control worms. There was also a difference between the two groups of females; the female worms recovered from *EBi3⁻/⁻*I were 9.1 times downregulated compared to the control female. It was also possible to observe difference between the two groups of males; the male worms recovered from *EBi3⁻/⁻*I were 7.4 times downregulated compared to the control male. The recovered *EBi3⁻/⁻*I male was 3.2 times upregulated compared to the recovered *EBi3⁻/⁻*I female.

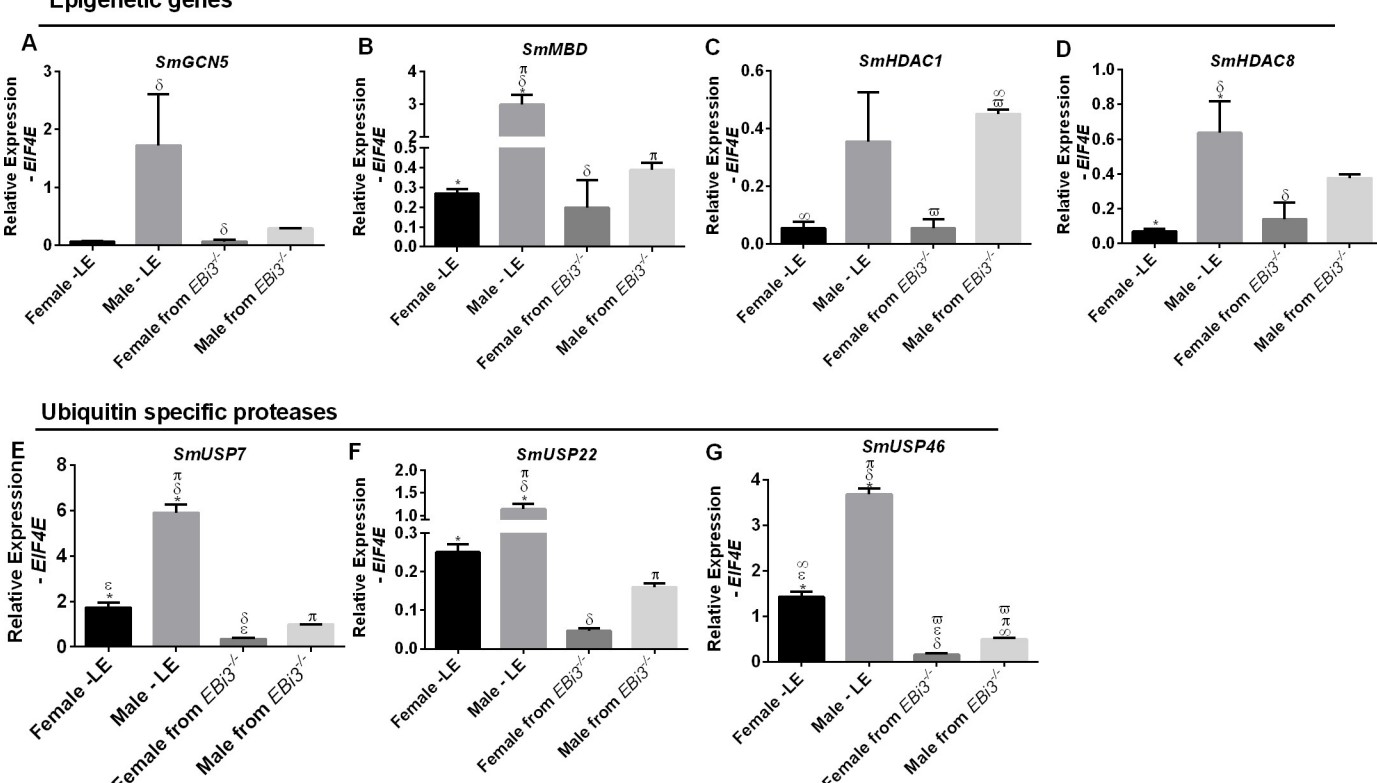

**Fig 6. Genes modulators of epigenetics: Expression in *S. mansoni*. (A)** Histone acetyltransferase, *SmGCN5*, p = 0.0084; **(B)** methyl binding domain, *SmMBD*, p < 0.0001; **(C)** histone deacetylase, *SmHDAC1*, p = 0.0002; and **(D)** *SmHDAC8*, p = 0.0027. Ubiquitin-specific protease expression in *S. mansoni*: **(E)** *SmUSP7*, p < 0.0001 **(F)** *SmUSP22* p < 0.0001, and **(G)** *SmUSP46*, p = 0.0087. Female and male LE represents parasites recovered from WT mice; female and male *EBi3⁻/⁻* represent parasites recovered from knockout mice; all mice were infected with 100 cercariae. Experiments were performed in triplicate using a pool of at least 50 parasites per extraction. One-way ANOVA, p < 0.01; GraphPad Prism 6. Comparisons indicated by the following symbols are significant. ϖ Female from *EBi3⁻/⁻* vs. male from *EBi3⁻/⁻*. π Male from LE vs. male from *EBi3⁻/⁻*. δ Male from LE vs. female from *EBi3⁻/⁻*. ∞ Female from LE vs. male from *EBi3⁻/⁻*. Ɛ Female from LE vs. female from *EBi3⁻/⁻*. * Female from LE vs. male from LE. Error bars show the standard deviation.

*SmGCN5* was differentially expressed between male control worms which were 24.4 times upregulated compared to the recovered *EBi3⁻/⁻*I female. Considering *SmMBD* expression, male control worms were upregulated 11.1 times compared to female control worms. The recovered male *EBi3⁻/⁻*I mouse was downregulated 7.7 times in relation to the control male. In the *SmHDAC1* gene the expression of the recovered *EBi3⁻/⁻*I female was 8.3 times downregulated compared to the recovered *EBi3⁻/⁻*I male. In the *SmHDAC8* gene male control worms were 9.3 times upregulated compared to female control worms.

## Discussion

At the beginning of the acute phase of schistosomiasis, the immune system of the definitive host aims mainly to attack the worms, initiating an intense Th1 response [24]. Oviposition activates the Th2 immune response, which aims to destroy the eggs; this is beneficial for the host since the Th1 response causes greater liver damage [25]. But induction of Th2 is also advantageous for the parasite, because it thus escapes the focus of the immune system. *S. mansoni* eggs produce and secrete a number of substances harmful to the liver, an example of which is the Omega 1 ribonuclease that is internalized by dendritic cells causing a decrease in the production of IL12, an important interleukin in the maintenance of Th1 [26]. We experimentally infected wild-type mice and *EBi3⁻/⁻* knockout mice, lacking functional IL27 and IL35

pathways, with *S. mansoni*. Similar numbers of parasites and pairs of parasites indicated that infection was equally efficient. However, the changes to the immunological background of mice induced a reduction in the number of eggs in feces and retained in the liver and also a reduction in granuloma number and volume.

Granulomas in *EBi3⁻ᐟ⁻*I mice seemed to be in the early stage, judging from their low quantity of extracellular matrix and predominance of disorganized inflammatory cells. Hepatic stellate cells (HSCs) are activated during the Th1 response, induced by secreted cytokines [27]. Activated HSCs produce and secrete extracellular matrix, IL27 and IL35 are interleukins belonging to the Th1 response [7,27]. In our knockout mice infected with *S. mansoni*, we observed predominately inflammatory cells in hepatic granuloma and few collagen fibers; this may have been due to the absence of the immune response performed by IL27 and IL35, then, the Th1 response in the liver might enhance the increase of fibrosis during infection by *S. mansoni*.

A previous study using a knockout model of the *WSX* chain of the IL27 receptor demonstrated a decrease in Th1 response-related cytokines without impacting changes in granuloma, using 80 cercariae of the Puerto Rican *S. mansoni* strain for infection [6]. In our model, 100 cercariae of the LE strain were used, and the volume and number of granulomas were reduced in *EBi3⁻ᐟ⁻* livers at the acute phase of schistosomiasis.

The DNA methylation and demethylation of liver affected by schistosomiasis remains unexplored. In our approach the S2 highlights the expression of all genes in each group facilitating the perception of patterns, for example, analyzing each group separately we can see tendencies to methylation of hepatic DNA during infection considering WTC versus WTI. *S. mansoni* infection led to an increase in DNA methylation in WTI mice, indicated by a decrease of statistical differences among relative expression levels of all *TET* and *DNMT* genes in the WTI group compared to statistical differences among *TET* and *DNMT* expression levels in WTC; comparing Fig 2 with S2 suggests that *DNMT3A* plays an important role in the rate changes observed. In *EBi3⁻ᐟ⁻* mice, infection caused an increase in DNA demethylation and analyzing *EBi3⁻ᐟ⁻*C versus *EBi3⁻ᐟ⁻*I we can see that *TET1* plays the major differences observed in S2, but *TET2* and *TET3* also increased strongly their expression values. The literature lacks data about methylation and demethylation of DNA in hepatic tissues during schistosomiasis, but considering other disease that affect the liver causing collagen deposition, there is an increase in liver methylation during hepatic fibrosis development, as determined by analyzing the overall methylation and the methylation of specific genes (Septina9, Long non-coding RNA H19, SUN2) in diseases such as hepatocarcinoma, hepatitis and cirrhosis [10,11,28], this occurred in our control groups WTC and WTI considering the qRT-PCR data.

MiRNAs are well explored in *S. mansoni* adult males and females have different repertoires or amounts of some miRNAs, many of which are important for the normal development of the parasite's reproductive organs in addition, these molecules are distributed for all chromosomes autosomal and sexual [29]. Adult worms have miRNAs with some predicted targets located in the definitive host, reinforcing the role of these molecules in parasite-host interaction [30]. A previous work of our group identified a set of miRNAs in silico highlighting their conservation in *S. mansoni* [31]. Recently some miRNAs present in the work of our group cited before were found in *S. mansoni* exosomes from parasite culture medium and also in mice serum, bouncing their potential to act in host targets and to be involved in host-parasite interactions [30]. Some of these miRNAs were tested in the experiments of this present work to detect if exists also alterations in miRNA expression influenced by immune response of the host, given its importance as possible biomarkers and interactions with the host. Our results showed statistical differences in expression of miR-190-3p and miR-190-5p between the fold change of males and females, reinforcing the importance of miRNAs for adaptation of the

parasite to the host biased by sex and coupled to immune response of the host. The miR-125A and miR-125-3p were more expressed in females worms recovered from WTI than females recovered from *EBi3⁻/⁻*I mice, males worms recovered from *EBi3⁻/⁻*I mice did not expressed, and so it was not possible to perform fold change to this group.

MiR-122 has been analyzed in the mice livers because it inhibits targets that promote STAT3 phosphorylation, and it has also been reported that B and T cell differentiation increases in the presence of EBi3 through the gp130 and STAT3 receptors. MiR-122 inhibits STAT3 indirectly, and this receptor is part of the response generated by EBi3 [32,33]. In this work it was observed that there is a negative correlation between this miRNA and the DNA methyltransferase and demethylase genes (*DNMT1* and *TET3*) and positive with the number of granulomas found in the liver of *EBi3⁻/⁻*I mice, which does not occur in WTI. This correlation was induced by knockout of the pathway, since this miRNA indirectly influences EBi3. In *EBi3⁻/⁻*C miR-122-3p also correlated with *DNMT1* expression, indicating that this association may be instrumental in the knockout model. In WTC miR-122-3p showed positive correlations with methylase and demethylase gene expressions (*DNMT3A* and *TET2*), the opposite occurred in infected knockouts and controls, no correlation was observed between miRNAs in the WTI group. The miRNAs that are part of the same pre-miRNA, miR-122-3p and 5p also correlated positively.

Of the *S. mansoni* USPs analyzed here, *SmUSP7* has the most domains, enabling the most functions, and this SmUSP therefore presents the highest level of expression independently of the strain [21]. *USP7* orthologs have the ability to deubiquitinate histone acetyltransferase Tip60, which is involved in the acetylation of various histone H2A subtypes, deubiquitinate lysine 63 preventing the degradation of histone deacetylase Sirt7, and deubiquitinate histone H3 in a process involving the methylation of newly synthesized DNA by *DNMT1* [34–36]. In our study of *SmUSP7* expression, control male parasites showed relative expression levels of between 6 and 8 while control female parasites presented relative expression levels of approximately 2.

*SmUSP15* was expressed at minimal levels. This enzyme deubiquitinates TGFβ receptors, preventing their degradation, and is also involved in Smad deubiquitination, altering their cellular localization and signaling [37]. *USP15* orthologs also promote H2B deubiquitination related to transcriptional activation [38]. *SmUSP22* expression levels were reduced in parasites recovered from knockout animals in relation to those in control parasites; knockout mice immunological system influenced in males parasites expression conducting them to downregulation. USP22 participates in the SAGA transcriptional activating complex, and its downregulation might be related to chromatin repression [39].

*SmUSP46* expression levels in parasites recovered from knockout animals were much lower than those in parasites recovered from wild-type mice, but proportional expression in males and females was similar. In other species, USP46 deubiquitinates deubiquitinate histones H2A and H2B associated to USP12 and WDR48 proteins, reinforcing its epigenetic function [40]. The paralog *SmUSP49-44* showed no significant changes in expression level in parasites recovered from knockout or wild-type mice. In other species, the USP49-44 enzyme deubiquitinates H2B histones [41]. Our results indicated that the host immune system does not cause changes in *SmUSP49-44* expression level.

*SmMBD* expression levels were lower in male parasites recovered from knockout animals than in those recovered from wild-type mice. This gene has been linked to normal egg production and number of eggs produced [20], and our *EBi3⁻/⁻*I mice eliminated smaller quantities of eggs in feces. *SmMBD* forms part of the NuRD repression complex along with the HDAC8 enzyme [42]. In our study, this machinery was downregulated in male parasites recovered from *EBi3⁻/⁻*I mice due to decreased *SmMBD* expression; *SmHDAC8* expression levels revealed

only a loss of sex-specific function in worms recovered from *EBi3*$^{-/-}$I mice. Motility and viability of schistosomula and adults may also be affected by the absence of histone deacetylase activity [43]. *SmHDAC1* expression levels revealed a notable difference between females recovered from WT mice and females recovered from *EBi3*$^{-/-}$ mice, and males and female parasites recovered from *EBi3*$^{-/-}$I mice gained sex-specific functions. Epigenetic plasticity is also important for fertility maintenance in the parasite, as demonstrated by the inhibition of histone acetyltransferase genes and enzymes [44]. The expression patterns of *SmHDAC3* did not vary under our experimental conditions.

Decreased oviposition and worm size has already been observed in other models of knockout, such as hepatic CD4$^{+}$ T-lymphocyte$^{-/-}$, in addition to decreased male-female pairing [45]. More recent studies have detected that nematodes are stimulated by the induction of some interleukins. The presence of IL 5 increased nematode reproduction and parasites recovered from IL 5$^{-/-}$ mice had larvae size smaller than the larvae size recovered from control, thus concluding that this interleukin promoted faster development and reproduction without altering other parasite characteristics [46]. Our EBi3 knockout model may have induced decreased worm oviposition. It has been described that p28 and p35 generate signaling independent of EBi3 protein, but this signaling has unknown function and impact so far [7]. In the study that assessed the impact of the WSX-1 knockout chain (the only receptor chain that is not shared with other interleukins besides IL27) the authors detected a decrease in IFNγ [6]. In this study, the absence of IL27 due to EBi3 knockout could also have triggered a decrease in IFNγ and a decrease in granulomas volume, since this cytokine is important in the acute phase of schistosomiasis. The impact of IL35 should be more evident in the chronic phase of schistosomiasis mansoni, in which there is an increase in the regulatory immune response, since IL35 has a strong anti-inflammatory/regulatory function. The host reacted to this softer pattern of infection and modulated the expression of genes responsible for alterations in silencing and activation in DNA, which may interfere with the transcriptional regulation of several genes (Fig 7).

In this study we analyzed a set of genes related to histone modifications and miRNAs in *S. mansoni* and evaluated the parasitological profile of infected EBi3 knockout mice. Altered miRNAs during hepatic fibrosis, DNA methylation/demethylation genes that influence gene regulation and the overall 5-mC content were analyzed in the definitive host. With these analyzes it was possible to realize that the parasite behaves differently in response to the host immune system. Chromatin compaction/decompression genes are altered in *S. mansoni*, this reflected in the amount of eggs produced that were detected in the liver and feces of mice. As we analyzed the expression of epigenetic genes in *S. mansoni*, the observed differences are likely to influence the expression of other parasite genes. Cercariae exhibit significant survival variability depending on the mouse strain used and schistosomiasis vaccines have so far shown little effectiveness [47]. Perhaps the ability of the parasite to adapt to the host immune system triggering a different infection profile as shown in this study is one of the factors that make this discovery difficult if we compare controls with knockouts.

## Methods

### Ethics statement

The studies were approved by CEUA—FMRP–USP (Ethics Committee on Animal Use–Medical College of Ribeirão Preto–University of São Paulo) and are in accordance with Ethic and Animal Use Committee of the National Council for Control of Animal Experimentation—CONCEA-FMRP USP protocol no. 195/2015, law No. 11,794, of October 8, 2008. This

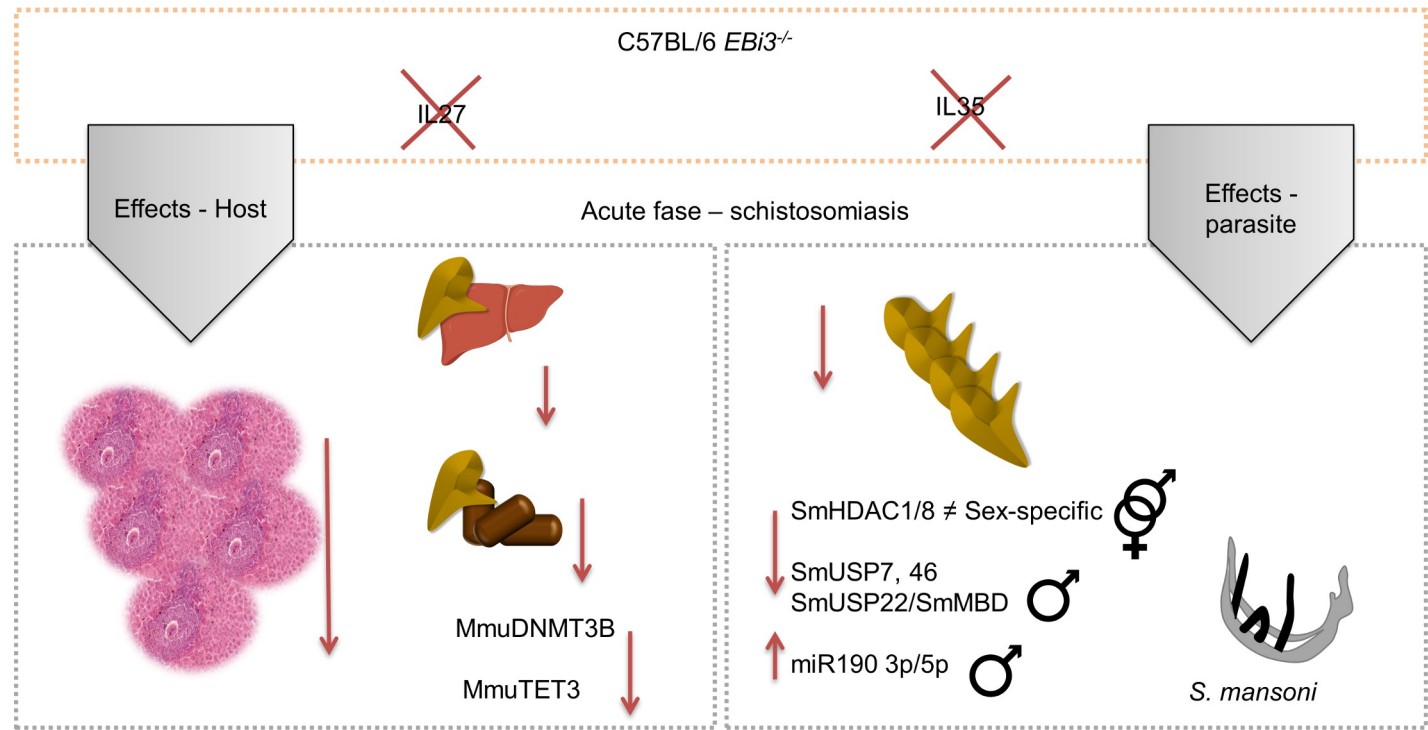

**Fig 7. Effect of the mouse *EBi3<sup>-/-</sup>* immune system on schistosomiasis.** Comparative scheme of the main immune system impacts on parasite and host. The immune system can modulate gene and miRNA expression in worms and the quantity of eggs they release resulting in less hepatic damage. In definitive host the number of granulomas, eggs/g of feces, expression of *TET3* were reduced, besides the increasing the % 5-mC in liver DNA.

protocol meets national and international standards of use and care. Knockout and wild mice were conditioned according to Quirino et al. [48].

## Parasites

*S. mansoni* (LE strain) parasites recovered from infected C57BL/6 mice (WTI) and infected C57BL/6 *EBi3<sup>-/-</sup>* mice (*EBi3<sup>-/-</sup>*I) were obtained by hepatic portal system perfusion of the animals, after 55 days of infection. Male and female parasites were counted and separated in plates and stored at -80°C (Fig 8).

## Livers

Mice *EBi3<sup>-/-</sup>* and WT had 17 g and 5 weeks of life when they were infected and remained 55 days infected, the uninfected groups *EBi3<sup>-/-</sup>*C and WTC waited the same time in the same conditions, only males were used for all experiments. When the mice were euthanized they had 13 weeks of life approximately. Livers from all mice were divided, left medial lobe for RNA extraction, left lateral lobe for parasitological tests (count of eggs if infected), right medial lobe for DNA extraction and right lateral lobe along with caudate process for microscopy. Thus, at the end of experiment was possible to have a result corresponding to each mouse separately. In this work were performed two perfusions, the first with 12 WTC, 10 WTI, 5 *EBi3<sup>-/-</sup>*C and 15 *EBi3<sup>-/-</sup>*I. The second perfusion were conducted only with knockout mice, 5 *EBi3<sup>-/-</sup>*C and 13 *EBi3<sup>-/-</sup>*I, to recover livers tissues to expression of lncRNAs, to increase the control knockout group and to rescue more worms. We did not acquire microscopic and parasitological parameter from the second perfusion. The correlations were analyzed using only the first perfusion

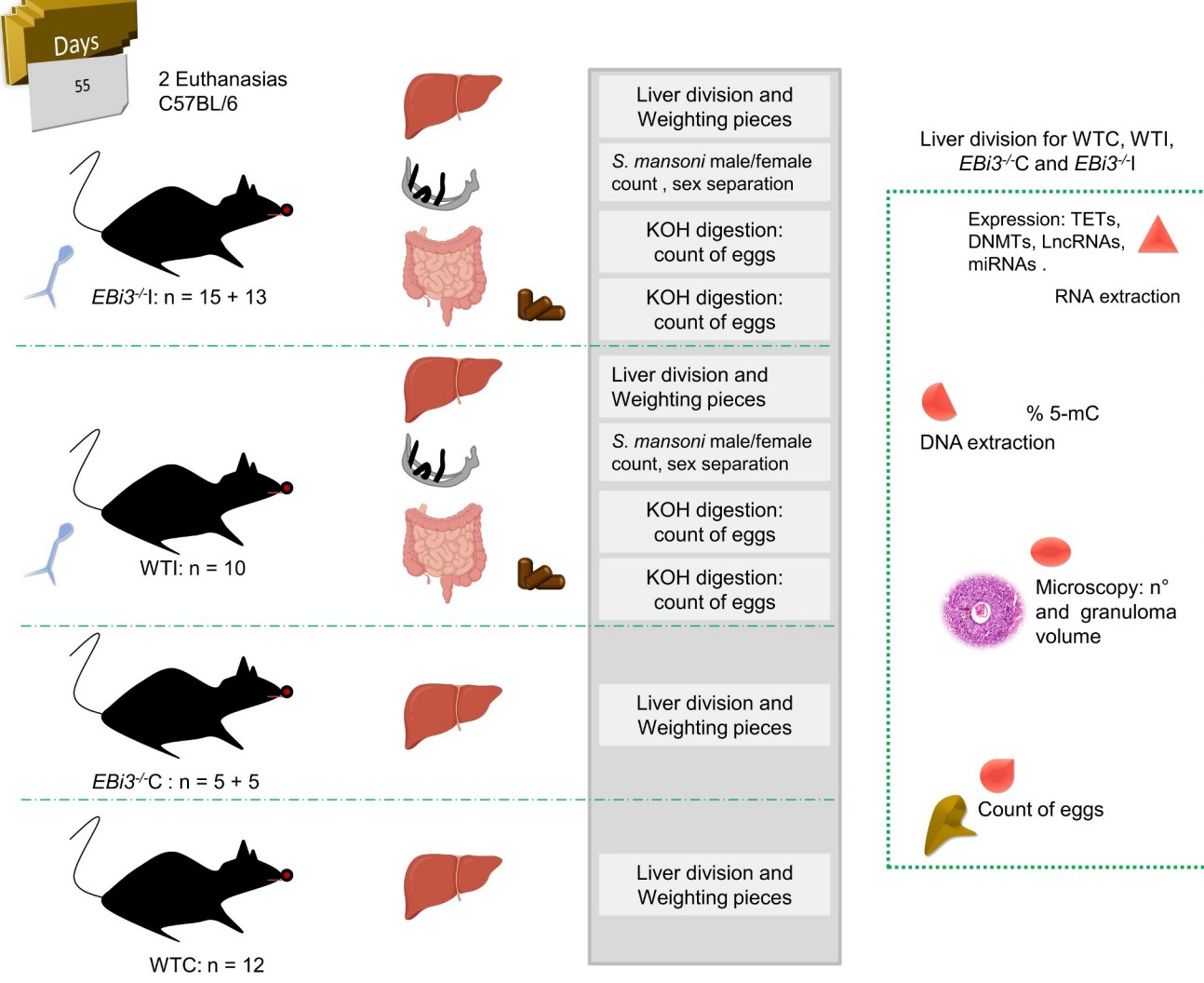

**Fig 8. Experimental scheme to compare molecular and parasitological parameters in mouse *EBi3*<sup>-/-</sup> and WT infected with *S. mansoni* and their influence on the male and female parasites.** Male mice *EBi3*<sup>-/-</sup> and WT had 17 g and 5 weeks of life when they were infected and remained 55 days infected, and the uninfected groups *EBi3*<sup>-/-</sup>C and WTC waited the same time in the same conditions. During the euthanasia, feces of the last day were collected from the cages. The couple worms collected were mechanically separated by sex, counted and stored; the single worms were counted and stored too. Intestines weighting and identification allow collect data individually. The division of livers into four parts, identification of each mouse part and weighting in order to obtain data more precise.

data for infected mice, aiming to plot the parameters corresponding to a given mouse properly in a row, (Fig 8).

## Parasitological parameters

Feces obtained the night before euthanasia were homogenized of each mouse separately in phosphate buffer (PBS) containing 10% formaldehyde until the day of counting; aliquots of 100 μL were used for counting with the aid of light microscopy. After euthanasia, mouse liver and intestine were weighted and after incubated separately in 10% KOH solution for 16 h in 15 mL tubes. Subsequently, the digested organs were centrifuged at 200 *g* for 2 min, washed in 0.85% NaCl solution and stored at 4˚C for counting; aliquots of 100 μL were used for counting with the aid of light microscopy.

After euthanasia of mice, one part of the liver was fixed in 1× PBS buffer supplemented with 10% formaldehyde for the preparation of hematoxylin and eosin-stained slides (HE); the slides were prepared in the NUPEB/UFOP Microscopy Multiuser Laboratory and examined using a Leica DM5000B microscope. Photodocumentation was established at a 20X magnification. The volume of granulomas was measured using granuloma radius in the ImageJ program (NHI), described in [49]. Statistical significance was calculated by *t* test in Graph Pad Prism 6; $p < 0.05$ was considered significant.

## *S. mansoni*: Expression of genes epigenetic modulators

Male and female *S. mansoni* LE worms recovered from WTI and *EBi3⁻/⁻*I mice were used for RNA extraction. Pools containing 50 worms were homogenized in 700 µL Tri Reagent (Sigma Aldrich, San Luis, Missouri, USA) using a Polytron (3 pulses, 15 s). DNA was degraded by three independent rounds of digestion RNase-free DNase I, and a Promega SV total kit (Promega, Madison, Wisconsin, USA) was used to clean RNA according to the manufacturer's instructions. RNA was quantified using a Nano Drop spectrophotometer (Thermo Fischer Scientific, Carlsbad, CA, USA) and stored at -80˚C. RNA integrity was evaluated in agarose/formaldehyde electrophoresis gels.

Aliquots containing 1 µg of total extracted RNA were used as template for first-strand cDNA synthesis (High Capacity RT-PCR System, Applied Biosystems), according to the manufacturer's recommendations in a thermocycler version 3.2 (Biocycler). cDNA was stored at -80˚C.

cDNA (100 ng) was added to oligonucleotides (2.5 µM) and Power SYBR Green Master Mix (1X), and samples were transferred to 96-well plates for the qPCR reaction. Gene amplification was performed in an Applied Biosystems ABI7300. Ultrapure water and total RNA were used as negative controls. Relative gene expression was quantified by the $2^{-\Delta Cq}$ method using *EIF4E* as an endogenous control (Table 1) [50]. All experiments were conducted in triplicate.

## Expression of *de novo* and maintenance DNA methyltransferases, demethylases and long non-coding RNAs in infected and uninfected wild-type and knockout mice

RNA was extracted from the livers of uninfected and infected C57BL/6 (WTC and WTI) and uninfected and infected C57BL/6 *EBi3⁻/⁻* (*EBi3⁻/⁻*C and *EBi3⁻/⁻*I) mice. About 100–120 mg of tissue was homogenized with 1 mL of Tri Reagent (Sigma Aldrich) with the aid of a Polytron, using 3 pulses of 30 s, RNA was extracted using chloroform and precipitated with ethanol. DNA was degraded with DNAse Turbo (Thermo Fisher) as previously described, and a Promega SV total kit was used to clean RNA according to the manufacturer's instructions. RNA was quantified using a Nano Drop spectrophotometer (Thermo Fisher) and stored to -80˚C. RNA integrity was evaluated in 1.2% agarose/formaldehyde electrophoresis gels. Total extracted RNA (1 µg) was used as template for first-strand cDNA synthesis (High Capacity RT-PCR System, Applied Biosystems), according to the manufacturer's recommendations using a thermocycler (Biocycler, version 3.2). cDNA (100 ng) was added to oligonucleotides (2.5 µM) and SYBR Green Select Master Mix. Ultrapure water and total RNA were used as negative controls. Relative gene expression was quantified by the $2^{-\Delta Cq}$ method using *HPRT* as an endogenous control (Table 2) [50]. We had a tube of RNA and other of cDNA corresponding to a mouse, totalizing 60 tubes, each gene analyzed had a technical triplicate, and we did not ever used pools for mice experiments.

## Liver and *S. mansoni* miRNA expression

miRNA extraction was conducted using 50 mg of liver from *EBi3$^{-/-}$*C and *EBi3$^{-/-}$*I mice or three replicates containing 50 male *S. mansoni* worms and three replicates containing 50 female worms. Extractions were performed using a miRNeasy kit (Qiagen) following the manufacturer's protocol in order to obtain a sample enriched with miRNAs. Each biological sample was homogenized in 700 μL QIAzol Lysis Reagent (Qiagen) with the aid of a Polytron-type homogenizer. Worm and liver samples were treated three times with Ambion DNAse Turbo (Thermo Fisher) to eliminate DNA contamination. After extraction, samples were quantified using a Thermo Fischer Nano Drop and the integrity of the samples was evaluated on 1.2% agarose/formaldehyde gel. Reverse transcription was performed using 1 μg of RNA with a miScript II cDNA (Qiagen) preparation kit RT for the evaluation of miRNA expression according to the manufacturer's recommendations. Gene expression was assayed using a miScript SYBR Green PCR kit for expression of liver miRNAs on an Applied Biosystems 7300 platform and quantified by the $2^{-\Delta Cq}$ method. Gene expression was assayed using a Taqman miRNA assay kit (Applied Biosystems, Foster, CA, USA) for expression of *S. mansoni* miRNAs on an Applied Biosystems 7500 platform and quantified by the $2^{-\Delta\Delta Cq}$ method. We conducted this experiment with parasite pools and as the expression of mice genes described before, we used 60 mice individually, 1 biological replicate for mouse for each analysis and three technical replicates to all miRNAs analyzed.

## DNA 5-mC % content in mouse liver

Liver genomic DNA from WTC, WTI, *EBi3$^{-/-}$*C and *EBi3$^{-/-}$*I mice was extracted from 30 mg of liver homogenized with liquid nitrogen in a crucible following the steps and reagents from the Wizard kit (Promega), this experiment were conducted with 42 mice, 1 biological replicate for each analysis from all groups of the first perfusion in technical duplicates. Samples were quantified by Qubit fluorometric quantitation (Thermo Fisher). The integrity of the extracted DNA was evaluated on 0.6% agarose gel. Quantification of 5-methylcytosine was performed using a 5-mC DNA Elisa kit, catalog number D5325 (Zymo Research, Irvine, CA, USA) with 100 ng of DNA input in duplicates according to the manufacturer's instructions.

## Statistical analysis

Parasitological profile of mice represented the animals of the first perfusion, WTC (n = 12), WTI (n = 10), *EBi3$^{-/-}$*C (n = 5) and *EBi3$^{-/-}$*I (n = 15). All parasitological parameters were analyzed with $p < 0.05$, for eggs on intestine the test used was Unpaired T with Welch's correction, we could not found significance. For Eggs on the liver, the test used was Unpaired T with Welch's correction, we found significance of $p = 0.0029$. For number of parasites per mouse, the test used was Unpaired T with Welch's correction, no significance was found. For number of couples per mouse, the test used was Unpaired T with Welch's correction, no significance was found. For eggs per gram of feces the test used was Unpaired T with Welch's correction, the significance of $p < 0.0001$. For number of granulomas, Mann-Whitney test presented significance of $p = 0.0001$. For granulomas volume, Unpaired T with Welch's correction presented significance of $p < 0.0001$.

Expression of TETS and DNMTs in mice liver was analyzed using Kruskal-Wallis test, $p < 0.05$. DMNT1 $p = 0.0348$; DNMT3A $p = 0.0008$; DNMT3B $p = 0.0001$; TET1 $p = 0.0041$; TET2 $p = 0.0001$ and TET3 $p = 0.0001$. Isolate analysis of each group also was conducted using Kruskal-Wallis test, $p < 0.05$. WTC $p < 0.0001$; WTI $p < 0.0001$; *EBi3$^{-/-}$*C $p = 0.0015$ and *EBi3$^{-/-}$*I $p < 0.0001$. These analysis represented the animals of the two perfusions, WTC (n = 12), WTI (n = 10), *EBi3$^{-/-}$*C (n = 10) and *EBi3$^{-/-}$*I (n = 28).

**Table 2. Oligonucleotides used to evaluate gene expression in C57BL/6 mouse liver.**

| Oligonucleotides | Sequence | Source |
|---|---|---|
| *MmuHPRT* GenBank (NC_000086.7) | F: 5'-TGACACTGGCAAAACAATGCA-3'<br>R: 5'-GGTCCTTTTCACCAGCAAGCT-3' | [53] |
| *MmuDNMT1* GenBank (NC_000075.6) | F: 5'-CTACCTGGCTAAAGTCAAGTC3-'<br>R: 5'-CACTCTCTGTGTCTACAACTC-3' | [54] |
| *MmuDNMT3A* GenBank (NC_000078.6) | F: 5'-GCACCTATGGGCTGCTGCGAAGACG-3'<br>R: 5'-CTGCCTCCAATCACCAGGTCGAATG-3' | [54] |
| *MmuDNMT3B* GenBank (NC_000068.7) | F: 5'-CAAGGAGGGCGACAACCGTCCATT-3'<br>R: 5'-TGTTGGACACGTCCGTGTAGTGAG-3' | [54] |
| *MmuTET1* GenBank (NC_000076.6) | F: 5'-GAGCCTGTTCCTCGATGTGG-3'<br>R: 5'-CAAACCCACCTGAGGCTGTT-3' | [55] |
| *MmuTET2* GenBank (NC_000069.6) | F: 5'-AACCTGGCTACTGTCATTGCTCCA-3'<br>R: 5'-ATGTTCTGCTGGTCTCTGTGGGAA-3' | [54] |
| *MmuTET3* GenBank (NC_000072.6) | F: 5'-TCCGGATTGAGAAGGTCATC-3'<br>R: 5'-CCAGGCCAGGATCAAGATAA-3' | [54] |

Spearman correlation was used in matrix and in dual correlations, the results represented the animals of the first perfusion except *EBi3<sup>-/-</sup>*C, WTC (n = 12), WTI (n = 10), *EBi3<sup>-/-</sup>*C (n = 10) and *EBi3<sup>-/-</sup>*I (n = 15), because all parameters were tested for the first perfusion in infected animals. *EBi3<sup>-/-</sup>*C was integrated in matrix because control mouse did not contain eggs on liver or intestine or in stool and the parasitological parameters were the only data that we did not obtain on second perfusion. For matrix correlations results, the p-values and coefficients are plotted in a table at supplemental material. Spearman correlation between number of granulomas and expression of TET3 were significant p = 0.0415, and also spearman correlation between number of granulomas and expression of DNMT1 p < 0.0001.

Anova one-way test analyzed the differences among treatments considering the expression of genes in *S. mansoni*, p < 0.01. The following genes were significant, *SmGCN5* p = 0.0084; *SmUSP7* p < 0.0001; *SmUSP22* p < 0.0001; *SmUSP46* p = 0.0087; *SmHDAC1* p = 0.0002; SmHDAC8 p = 0.0027; and *SmMBD* p < 0.0001.

*S. mansoni* miRNAs expression were significant to miR-190-3p (p = 0.0055) and 5p (p = 0.0033), using Unpaired T with Welch's correction, p < 0.01. Expression of liver miRNAs was analyzed Mann–Whitney U tests, p < 0.05. These analysis represented the animals of the two perfusions, WTC (n = 12), WTI (n = 10), *EBi3<sup>-/-</sup>*C (n = 10) and *EBi3<sup>-/-</sup>*I (n = 28).

Percentages of 5-mC DNA of liver mice samples represented the animals of the first perfusion, WTC (n = 12), WTI (n = 10), *EBi3<sup>-/-</sup>*C (n = 5) and *EBi3<sup>-/-</sup>*I (n = 15). Anova one-way test was performed, p < 0.05, the significance of results valued p = 0.0004.

## List of accession numbers/ID numbers for genes and proteins mentioned in the text

Gene sequences accessible on GenBank: NC_000086.7 *MmuHPRT;* NC_000075.6 *MmuDNMT1;* NC_000078.6 *MmuDNMT3A;* NC_000068.7 *MmuDNMT3B;* NC_000076.6 *MmuTET1;* NC_000069.6 *MmuTET2;* NC_000072.6 *MmuTET3;* NC_000083.6 Rnu6.

Gene sequences accessible on GeneDB: Smp_070190 *SmGCN5;* Smp_005210 *SmHDAC1;* Smp_093280 *SmHDAC3;* Smp_091990 *SmHDAC8;* Smp_089180 *SmUSP 7;* Smp_128770 *SmUSP 15;* Smp_074400 *SmUSP 22;* Smp_000710 *SmUSP 46;* Smp_123630 *SmUSP 49/44;* Smp_138180 *SmMBD;* Smp_001500 *SmEIF4E.*

LncRNAs transcript ID: TCONS_00001011; TCONS_00012347; TCONS_00013257; TCONS_00003004; TCONS_00000625; TCONS_00001840; TCONS_00009100;

TCONS_00009852; TCONS_00009849; TCONS_00009851; TCONS_00012478; TCONS_00010393; TCONS_00010903; TCONS_00011021; TCONS_00013835.

MicroRNAs accessible on miRBase, *S. mansoni* miRNAs: MI0021819 miR-190-3p and miR-190-5p; MI0010676 miR-125A and miR-125-3p; MI0021818 miR-124-3p. Mice miRNAs: MI0000579 miR-31-3p and miR-31-5p; MI0000256 miR-122-3p and miR-122-5p.

## Supporting information

**S1 Fig. Appearance of granulomas in livers of wild-type and knockout mice. (A)** Wild-type uninfected; **(B)** *EBi3*[-/-] uninfected; **(C)** wild-type infected; **(D)** *EBi3*[-/-] infected. Slides prepared after 55 days of infection; livers were fixed in 10%formaldehyde and stained using Hematoxylin eosin.
(TIF)

**S2 Fig. Comparison between methylation and demethylation. (A)** WTC; **(B)** WTI; **(C)** *EBi3*[-/-]C; **(D)** *EBi3*[-/-]I. *HPRT* was used as constitutive gene control. Kruskal–Wallis, p < 0.05; GraphPad Prism6. Comparisons indicated by the following letters are significant. a: *DNMT1* vs. *DNMT3A*; b: *DNMT1* vs. *DNMT3B*; c: *DNMT1* vs. *TET1*; d: *DNMT1* vs. *TET2*; e: *DNMT1* vs. *TET3*; f: *DNMT3A* vs. *DNMT3B*; g: *DNMT3A* vs. *TET1*; H: *DNMT3A* vs. *TET2*; i: *DNMT3A* vs. *TET3*; j: *DNMT3B* vs. *TET1*; k: *DNMT3B* vs. *TET2*; L: *DNMT3B* vs. *TET3*; m: *TET1* vs. *TET2*; n: *TET1* vs. *TET3*; *TET2* vs. *TET3*. Error bars show the standard deviation.
(TIF)

**S3 Fig. Expression levels of USP genes *15* and *49/44* and *HDAC3* in *S. mansoni*.** Female and male LE represent parasites recovered from WT mice; female and male *EBi3*[-/-] represent parasites recovered from knockout mice; all mice were infected with 100 cercariae. Experiments were done in triplicate using pools of at least 50 parasites per extraction. One-way ANOVA, p < 0.01; GraphPad Prism 6. No alterations in the expression levels of these genes were observed among the groups under our experimental conditions. Error bars show the standard deviation.
(TIF)

**S1 Table. Correlation matrix: Several parameters in the same infected mice are correlated among themselves.** Spearman correlation, p < 0.05.
(XLSX)

## Acknowledgments

We would like to thank Olinda Mara Brigato and Wander Cosme Ribeiro da Silva for assisting in caring for the mice.

## Author Contributions

**Formal analysis:** Ester Alves Mota.

**Funding acquisition:** Renata Guerra-Sá.

**Investigation:** Ester Alves Mota.

**Methodology:** Ester Alves Mota, Andressa Barban do Patrocínio.

**Project administration:** Renata Guerra-Sá.

**Resources:** Vanderlei Rodrigues, João Santana da Silva, Vanessa Carregaro Pereira, Renata Guerra-Sá.

**Supervision:** Renata Guerra-Sá.

**Validation:** Ester Alves Mota.

**Visualization:** Ester Alves Mota.

**Writing – original draft:** Ester Alves Mota.

**Writing – review & editing:** Renata Guerra-Sá.

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
