## [Decision Letter · Decision Letter 0]

25 Oct 2019

Dear Dr Sa:

Thank you very much for submitting your manuscript "Epigenetic and parasitological parameters are modulated in EBi3-/- mice infected with Schistosoma mansoni" (#PNTD-D-19-01459) for review by PLOS Neglected Tropical Diseases. Your manuscript was fully evaluated at the editorial level and by independent peer reviewers. The reviewers appreciated the attention to an important problem, but raised some substantial concerns about the manuscript as it currently stands. These issues must be addressed before we would be willing to consider a revised version of your study. We cannot, of course, promise publication at that time.

We therefore ask you to modify the manuscript according to the review recommendations before we can consider your manuscript for acceptance. Your revisions should address the specific points made by each reviewer. 

When you are ready to resubmit, please be prepared to upload the following:

(1) A letter containing a detailed list of your responses to the review comments and a description of the changes you have made in the manuscript.

(2) Two versions of the manuscript: one with either highlights or tracked changes denoting where the text has been changed (uploaded as a "Revised Article with Changes Highlighted" file); the other a clean version (uploaded as the article file).

(3) If available, a striking still image (a new image if one is available or an existing one from within your manuscript). If your manuscript is accepted for publication, this image may be featured on our website. Images should ideally be high resolution, eye-catching, single panel images; where one is available, please use 'add file' at the time of resubmission and select 'striking image' as the file type. 

Please provide a short caption, including credits, uploaded as a separate "Other" file. If your image is from someone other than yourself, please ensure that the artist has read and agreed to the terms and conditions of the Creative Commons Attribution License at http://journals.plos.org/plosntds/s/content-license (NOTE: we cannot publish copyrighted images). 

(4) If applicable, we encourage you to add a list of accession numbers/ID numbers for genes and proteins mentioned in the text (these should be listed as a paragraph at the end of the manuscript). You can supply accession numbers for any database, so long as the database is publicly accessible and stable. Examples include LocusLink and SwissProt.

(5) To enhance the reproducibility of your results, we recommend that you deposit your laboratory protocols in protocols.io, where a protocol can be assigned its own identifier (DOI) such that it can be cited independently in the future. For instructions see http://journals.plos.org/plosntds/s/submission-guidelines#loc-methods

While revising your submission, please upload your figure files to the Preflight Analysis and Conversion Engine (PACE) digital diagnostic tool, https://pacev2.apexcovantage.com/ PACE helps ensure that figures meet PLOS requirements. To use PACE, you must first register as a user. Then, login and navigate to the UPLOAD tab, where you will find detailed instructions on how to use the tool. If you encounter any issues or have any questions when using PACE, please email us at figures@plos.org.

We hope to receive your revised manuscript by Dec 24 2019 11:59PM. If you anticipate any delay in its return, we ask that you let us know the expected resubmission date by replying to this email.

To submit a revision, go to https://www.editorialmanager.com/pntd/ and log in as an Author. You will see a menu item call Submission Needing Revision. You will find your submission record there. 

Sincerely,

Jennifer A. Downs, M.D., Ph.D.

Guest Editor

Timothy Geary

Deputy Editor

Reviewer's Responses to Questions

**Key Review Criteria Required for Acceptance?**

**Methods**

-Are the objectives of the study clearly articulated with a clear testable hypothesis stated?

-Is the study design appropriate to address the stated objectives?

-Is the population clearly described and appropriate for the hypothesis being tested?

-Is the sample size sufficient to ensure adequate power to address the hypothesis being tested?

-Were correct statistical analysis used to support conclusions?

-Are there concerns about ethical or regulatory requirements being met?

Reviewer #1: -Are the objectives of the study clearly articulated with a clear testable hypothesis stated?

The hypothesis is stated but should be made more clear.

-Is the study design appropriate to address the stated objectives?

yes

-Is the population clearly described and appropriate for the hypothesis being tested?

yes

-Is the sample size sufficient to ensure adequate power to address the hypothesis being tested?

yes

-Were correct statistical analysis used to support conclusions?

yes

-Are there concerns about ethical or regulatory requirements being met?

yes

Reviewer #2: The study design to address the stated objectives is poorly described. In particular the number of control mice (WTC, EBi3-/-C) and the number of infected mice (WTI and EBi3-/-I) used in this study is not specified, as well as the sex and age of the animals. A single infection experiment was performed in mice? 

The RNA for RT-qPCR studies is prepared from pooled samples. It is not specified the number of mice used for the preparation of RNA. Regarding the correlation studies the statistical significance of the data should be also indicated in the Figure Legends.

**Results**

-Does the analysis presented match the analysis plan?

-Are the results clearly and completely presented?

-Are the figures (Tables, Images) of sufficient quality for clarity?

Reviewer #1: -Does the analysis presented match the analysis plan?

yes, but the results are difficult to follow

-Are the results clearly and completely presented?

there are parts missing and presentation should be made clearer 

-Are the figures (Tables, Images) of sufficient quality for clarity?

must be improved

Reviewer #2: The text of the Results section is poorly presented. It could be helpful for readers to be guided in the list of genes that have been studied: i.e. criteria of selection, relevance. A more focused presentation on the most relevant gene expression variations could improve the quality of the manuscript. Maybe some data could be moved to the Supplemental data section.

Specific comments:

1. The data presented in Figure 1 are very interesting. The following information are not indicated: the numbers of control and infected mice used for the assessment of parasitological parameters; the number of independent infection experiments eventually performed; the sex and age of the mice used These informations should be included in Figure Legend and in the Experimental section.

2. Changes in de novo DNA methytransferase and DNA demethylase expression in

EBi3-/- linked to S. mansoni infection. Please specify that the analysis has been made on liver in the title. The significance is Kruskal–Wallis, p < 0.05 for all letters indicated? What each point represents? 

Please indicate in the caption with kind of significance in the correlation has been observed. Spearman correlations: please describe better the parameters that you are correlating: granuloma volume as stated in legend or granuloma number as indicated in (G) and (H). Please specify in Figure 2 caption, which value of significance in the correlation has been detected.

3. Please indicate in each Figure legend what each point represents (sample/mouse/technical replicate/biological replicate.

**Conclusions**

-Are the conclusions supported by the data presented?

-Are the limitations of analysis clearly described?

-Do the authors discuss how these data can be helpful to advance our understanding of the topic under study?

-Is public health relevance addressed?

Reviewer #1: -Are the conclusions supported by the data presented?

yes. But presentation and logics could be improved

-Are the limitations of analysis clearly described?

no

-Do the authors discuss how these data can be helpful to advance our understanding of the topic under study?

yes

-Is public health relevance addressed?

implicitly

Reviewer #2: The manuscript lack Conclusions and the authors do no discuss how these data can be used for further investigation on the role of the host immune system on schistosomiasis.

**Editorial and Data Presentation Modifications?**

Reviewer #1: (No Response)

Reviewer #2: Dear Editor,

In my opinion this manuscript needs a major revision. There are some interesting results but these are poorly presented.

**Summary and General Comments**

Reviewer #1: Motta et al. describe a very interesting phenomenon and the subject is definitely worth publishing. However, the matter of mutual influences of epigenetic information is complex and the manuscript was in this regard a bit difficult to follow. Essentially, I was often lost in my understanding if the authors talk about Schistosoma or mouse gene expression, RNA presence, modifications of DNA methylation etc.

Please make sure that the reader always knows where he is (in the parasite or in the host) and what the origin of the molecular is (parasite or host). This applies also to the discussion. Please do an effort to make clear where we are and who we talk about, and differentiate it also from the general case/assumption (e.g. line 367 I suppose it is meant that “in general HDAC8 shows nuclear and cytoplasmic location...”

DNA methylation is used equivalent to 5-methyl-cytosine, but there are more methylation modifications than this. Make this clear from the beginning or change terminology.

I find “epigene” a bit strange and it took me a while to understand that genes coding for chromatin modifying enzymes and chromatin associated RNA is meant.

I recommend that these general points and the specific points below should be addressed by the authors before further review.

Specific comments 

Line 19: “genomic imprinting” is a reserved term for imprinted genes. Don’t use it in your context.

Line 21: “could occur” since it’s a hypothesis 

In abstract and author summary explain what epigenetics (and stick to this definition throughout the manuscript) is and why it was investigated. Develop in more detail in the introduction.

Line 82: not clear why there is a link between sentence finishing by “...adaptation to the host.” and the next one “Among other factors...”

Line 87: “insert” should read “influence”?

Line 103: don’t use “imprinting”

Line 104-105: don’t understand the sentence 

Line 108: mention here which are the parasitic parameters that change

Line 149: in which tissue?

Line 180: talking about demethylating enzyme activity or 5mC level?

Line 196: 0.5% of what?

Line 194-202: good to know that the increase is statistically significant but what is the extent? Fold change or % difference with SD

Line 292: should this read “BUT induction of Th2...”?

Line 302: should “due to” read “judging from” or similar?

Line 317-330: I don’t get the point the authors want to put forward here.

Line 377: should read “…for the methylation and demethylation in the host(?),…” but then why methylation AND demethylation?

Line 380: the figure is not well drawn and as confusing as some parts of the manuscript. It is not clear what is host and what is parasite. Maybe colour code this and make the figure legend more clear.

Reviewer #2: The authors used C57BL/6 and C57BL/6 EBi3-/- murine (Mus musculus) models of S. mansoni infection in order to investigate possible changes in new and maintenance DNA methylation profiles in the liver after 55 days of infection. Moreover, the authors evaluated the expression levels of epigenetic genes and genes linked to histone deubiquitination in male and female S. mansoni worms by RT-qPCR. The subject is novel and of significance in the Schistosomiasis field and in particular in the host-parasite interplay. 

The data are poorly presented but the Manuscript can be improved. The most relevant findings should be highlighted. Importantly, the criteria used for the selection of genes to be investigated are not clearly presented. In the introduction an overview of the genes investigated in the paper should be included. Some information that are in the discussion section should be in the introduction to better follow the data presented in the Results section. The paper lack any Conclusion.

PLOS authors have the option to publish the peer review history of their article (what does this mean?). If published, this will include your full peer review and any attached files.

Reviewer #1: Yes: Christoph Grunau

Reviewer #2: No

---

## [Decision Letter · Decision Letter 1]

22 Jan 2020

Dear Dr Guerra-Sá,

We are pleased to inform you that your manuscript 'Epigenetic and parasitological parameters are modulated in EBi3-/- mice infected with Schistosoma mansoni' has been provisionally accepted for publication in PLOS Neglected Tropical Diseases.

Before your manuscript can be formally accepted you will need to complete some formatting changes, which you will receive in a follow up email. A member of our team will be in touch within two working days with a set of requests.

Best regards,

Jennifer A. Downs, M.D., Ph.D.

Guest Editor

Timothy Geary

Deputy Editor

Reviewer's Responses to Questions

**Key Review Criteria Required for Acceptance?**

**Methods**

-Are the objectives of the study clearly articulated with a clear testable hypothesis stated?

-Is the study design appropriate to address the stated objectives?

-Is the population clearly described and appropriate for the hypothesis being tested?

-Is the sample size sufficient to ensure adequate power to address the hypothesis being tested?

-Were correct statistical analysis used to support conclusions?

-Are there concerns about ethical or regulatory requirements being met?

Reviewer #2: YES

The manuscript has been revised accordingly to the reviewers' comments

**Results**

-Does the analysis presented match the analysis plan?

-Are the results clearly and completely presented?

-Are the figures (Tables, Images) of sufficient quality for clarity?

Reviewer #2: YES

The manuscript has been revised accordingly to the reviewers' comments

**Conclusions**

-Are the conclusions supported by the data presented?

-Are the limitations of analysis clearly described?

-Do the authors discuss how these data can be helpful to advance our understanding of the topic under study?

-Is public health relevance addressed?

Reviewer #2: YES

The manuscript has been revised accordingly to the reviewers' comments

**Editorial and Data Presentation Modifications?**

Reviewer #2: (No Response)

**Summary and General Comments**

Reviewer #2: The manuscript has been revised accordingly to the reviewers' comments.

PLOS authors have the option to publish the peer review history of their article (what does this mean?). If published, this will include your full peer review and any attached files.

Reviewer #2: No

---

## [Editor Report · Acceptance letter]

11 Feb 2020

Dear Dr Guerra-Sá,

We are delighted to inform you that your manuscript, "Epigenetic and parasitological parameters are modulated in EBi3-/- mice infected with Schistosoma mansoni," has been formally accepted for publication in PLOS Neglected Tropical Diseases.

Best regards,

Serap Aksoy

Editor-in-Chief

Shaden Kamhawi

Editor-in-Chief
